# AGENTRACER: WHO IS INDUCING FAILURE IN THE LLM AGENTIC SYSTEMS?

**Guibin Zhang**🦁†, **Junhao Wang**🐓†, **Junjie Chen**🦁†, **Wangchunshu Zhou**🌲,
**Kun Wang**🦌✉, **Shuicheng Yan**🦁✉
🦁NUS  🐓CUHK  🌲OPPO  🦌NTU
✉ Main Contact: `guibinz@outlook.com`

## ABSTRACT

Large Language Model (LLM)-based agentic systems, often comprising multiple models, complex tool invocations, and orchestration protocols, substantially outperform monolithic agents. Yet this very sophistication amplifies their fragility, making them more prone to system failure. Pinpointing the specific agent or step responsible for an error within long execution traces defines the task of **agentic system failure attribution**. Current state-of-the-art reasoning LLMs, however, remain strikingly inadequate for this challenge, with accuracy generally below 10%. To address this gap, we propose `AgenTracer`, the first automated framework for annotating failed multi-agent trajectories via counterfactual replay and programmed fault injection, producing the curated dataset `TracerTraj`. Leveraging this resource, we develop `AgenTracer-8B`, a lightweight failure tracer trained with multi-granular reinforcement learning, capable of efficiently diagnosing errors in verbose multi-agent interactions. On Who&When benchmark, `AgenTracer-8B` outperforms giant proprietary LLMs like Gemini-2.5-Pro and Claude-4-Sonnet by up 18.18%, setting a new standard in LLM agentic failure attribution. More importantly, `AgenTracer-8B` delivers actionable feedback to off-the-shelf multi-agent systems like MetaGPT and MaAS with $4.8 \sim 14.2\%$ performance gains, empowering self-correcting and self-evolving agentic AI. Codes are available at `https://github.com/bingreeky/AgenTracer`.

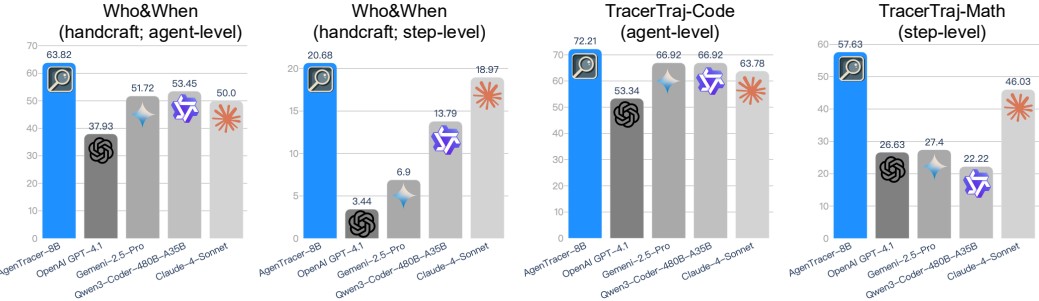

Figure 1: Benchmark performance comparison between `AgenTracer-8B` and leading industry providers.

## 1 INTRODUCTION

Large Language Model (LLM)-powered agents have exhibited exceptional proficiency across a wide array of cognitive faculties, encompassing perception (Driess et al., 2023; Wang et al., 2024; Zheng et al., 2023; Wei et al., 2024), planning (Zhu et al., 2024; Huang et al., 2024; Erdogan et al., 2025), reasoning (Putta et al., 2024; Masterman et al., 2024), and action (Li et al., 2024; Yang et al., 2024). As endeavors to address increasingly intricate challenges, such as iterative tool calling, complex document analysis, and multi-step web navigation, continue to surge, the inherent limitations of relying on a single monolithic model have become increasingly conspicuous.

Consequently, numerous multi-agent frameworks, motivated by collective intelligence (Wang et al., 2025a; Zhang et al., 2024a;b) and the society of mind theory (Minsky, 1988; Li et al., 2023), have emerged, ensembling multiple LLM agents to achieve refined subtask orchestration (Zhang et al.,

2025d; Hu et al., 2025), ultra-long context handling (Zhang et al., 2024d), and broader environmental perception (Jiang et al., 2024). These systems have demonstrated superior performance over single-agent counterparts across a range of complex real-world domains, including data science engineering (Hong et al., 2024), scientific discovery (Ghareeb et al., 2025; Ghafarollahi & Buehler, 2024), and software engineering (Wei et al., 2025). However, the integration of multiple autonomous agents alongside diverse auxiliary modules (*e.g.*, external databases, tools, and memory modules) inevitably exacerbates **system fragility**. As evidence, a recent empirical study by UC Berkeley (Cemri et al., 2025) reveals alarmingly high failure rates (reaching up to $86.7\%$) in popular multi-agent frameworks such as OpenHands (Wang et al., 2025b) and MetaGPT (Hong et al., 2023), with failure modes ranging from improper task decomposition to role disobedience. Such a high incidence of failure casts a long shadow over the real-world reliability of multi-agent systems.

A natural response to this challenge is **failure attribution**, *i.e.*, the precise identification of faulty components within the system upon task failure. Accurate failure attribution plays a critical role across multiple dimensions: **(I) system debugging**, by enabling agentic systems to perform more effective self-debugging across iterative attempts, thereby enhancing performance; **(II) data efficiency**, by leveraging failed agent trajectories to construct more informative training data; **(III) grounded self-improvement**, by facilitating grounded self-correction through precise agent credit assignment. Despite its importance, this process is predominantly left as a manual endeavor, requiring considerable human effort to analyze verbose trajectory logs.

While preliminary attempts have been made toward automation, their efficacy remains limited. For instance, Zhang et al. (2025c) employed state-of-the-art models such as OpenAI-o1 and DeepSeek-R1 (DeepSeek-AI et al., 2024) to perform failure attribution over trajectories from GAIA (Mialon et al., 2023), yet achieved accuracy below $10\%$. More critically, existing benchmarks for agentic failure attribution remain notably constrained. To the best of our knowledge, MAST (Cemri et al., 2025) and Who&When (Zhang et al., 2025c) contain only 200 and 127 manually annotated multi-agent failure trajectories, respectively, offering limited scale for systematic evaluation. Accordingly, substantial research gaps remain along two critical axes: ❶ **training resources**, concerning the automated construction of large-scale annotated multi-agent trajectories; and ❷ **methodology**, on developing swift and accurate multi-agent failure localizer, which we also refer to as a *tracer*.

**Present Framework.** To address the aforementioned research gap, we present **(1)** `AgenTracer`, a fully automated pipeline for constructing well-annotated multi-agent failure trajectories, and **(2)** `AgenTracer`-8B, a lightweight failure attributor for multi-agent systems. Methodologically, `AgenTracer` leverages ♣ **counterfactual replay**, systematically replacing agent actions with oracle guidance to identify the decisive error step responsible for system failure. To enhance data diversity and ensure annotation precision, it further adopts ♠ **programmatic fault injection**, synthetically generating failure instances by perturbing successful trajectories through targeted corruption.

Further, we fine-tune QWEN3-8B on the curated dataset to obtain `AgenTracer`-8B via reinforcement learning (RL), enabling it to analyze long-horizon multi-agent collaboration traces. `AgenTracer`-8B takes as input the trajectory log and associated environmental feedback, and outputs the identified decisive error step. A **multi-granular reward** is designed to supervise RL training, emphasizing both *step-level* and *agent-level* attribution accuracy. At inference, `AgenTracer`-8B enables malfunctioning agentic systems to rapidly identify the critical failure step along with explanations, facilitating automated multi-agent debugging. Our contributions are as follows:

❶ **Automated Pipeline.** We propose `AgenTracer`, the first automated pipeline for annotating multi-agent system failures, curating over 2,000 high-fidelity failure trajectories across seven datasets through counterfactual replay and programmatic fault injection.

❷ **Failure Tracer.** We develop `AgenTracer`-8B, a lightweight failure tracer dedicated for LLM agentic systems, trained via a multi-granular reinforcement learning to ensure accurate attribution at both the step and agent levels, facilitating automatic agentic system debugging.

❸ **Empirical Evaluation.** Experiments show that `AgenTracer`-8B facilitates *(I) accurate attribution*, outperforming giant LLMs like DEEPSEEK-R1 by $\sim 12.21\%$ and GEMINI-2.5-PRO by $\sim 18.18\%$ on Who&When benchmark; and *(II) self-evolution*, enabling off-the-shelf agentic systems to improve performance by $4.8 \sim 14.2\%$.

## 2 RELATED WORKS

**LLM-based Multi-Agent System.** Contemporary multi-agent systems can be broadly categorized by their level of automation into three classes: ■ **Handcrafted**, where the entire system configuration (*e.g.*, LLM backbone, prompting strategies, and communication protocols) is manually specified as in (Wu et al., 2023; Richards & et al., 2023; Li et al., 2023; Qian et al., 2023); ■ **Partially Automated**, which automate specific system components: for example, AutoAgent (Chen et al., 2023a), LLMSelector (Chen et al., 2025), and MasRouter (Yue et al., 2025) automate agent role assignment; DsPy (Khattab et al., 2023) and TextGrad (Yuksekgonul et al., 2024) optimize prompt design; GPTSwarm (Zhuge et al., 2024) and G-Designer (Zhang et al., 2024b) adaptively construct inter-agent topologies; ■ **Fully Automated**, where all modules within the system are autonomously designed and evolved (Hu et al., 2024a; Zhang et al., 2024c; 2025b; Wu et al., 2025; Nie et al., 2025; Gao et al., 2025; Zhang et al., 2025a). Our method is designed to provide precise error tracing across this entire spectrum of agentic systems. Accordingly, we curate trajectories from all three categories.

**Failure Attribution for Agents.** As multi-agent systems become increasingly intricate, incorporating multiple intelligent agents (Wang et al., 2023; Chen et al., 2023b), tool integrations (Shen et al., 2024), and communication protocols Marro et al. (2024), the resulting high error rates and structural fragility have emerged as critical concerns. MAST (Cemri et al., 2025) was the first to alarmingly characterize this issue, identifying fourteen prevalent failure patterns ranging from task disobedience to reasoning-action mismatches. Building upon this, Who&When (Zhang et al., 2025c) explored the feasibility of automating failure attribution, only to reveal that even giant reasoning models like DeepSeek-R1 fail catastrophically. `AgenTracer` advances the field by introducing both a scalable data synthesis pipeline and a lightweight, high-accuracy failure attributor.

**LLM-as-a-Judge & Credit Assignment.** Two other research topics closely related to this work are: **(I) LLM-as-a-Judge (LaaJ)**, which leverages LLMs/agents as evaluators based on pre-defined criteria for tasks such as data annotation (Latif et al., 2025; OpenAI, 2022), value alignment (Ji et al., 2025), reward modeling (Mahan et al., 2024; Lambert et al., 2024), and synthetic data generation (Sengupta et al., 2025; Hu et al., 2025). However, when applied to multi-LLM systems, LaaJ has shown limited effectiveness, as demonstrated in Zhang et al. (2025c); **(II) Credit Assignment** is a long-standing topic in multi-agent reinforcement learning (MARL), linking individual actions to long-term outcomes (Pignatelli et al., 2023; Arjona-Medina et al., 2019; Yin et al., 2023). Yet it remains largely unexplored in LLM agentic systems. The closest prior work, CollabUIAgents (He et al., 2025), uses LLMs to produce binary rewards per interaction, which are inherently unreliable. In contrast, `AgenTracer` achieves grounded credit assignment implicitly via principled attribution.

## 3 PRELIMINARY

In this section, we provide a general definition of LLM-based multi-agent systems and their operational workflow, and then formally define the objective of the multi-agent failure attribution.

**Notations.** Consider a LLM-based multi-agent system $\mathcal{M}$, tasked with collaboratively resolving a user-issued query $\mathcal{Q}$. The system consists of $N$ agents, indexed by $\mathcal{I} = \{1, 2, \ldots, N\}$, operating in discrete time under a turn-based protocol, *i.e.*, only one agent is active at each time step. Formally:

$$\mathcal{M} = \langle \mathcal{I}, \mathcal{S}, \mathcal{A}, \Psi, \mu \rangle, \tag{1}$$

where $\mathcal{S}$ denotes the set of system states; $\mathcal{A}$ is the overall action space, with each agent $i \in \mathcal{I}$ having a local space $\mathcal{A}_i \subseteq \mathcal{A}$; $\mu(t) \in \mathcal{I}$ specifies the agent scheduled to act at time $t$; $\Psi(s_{t+1} \mid s_t, a_t, \mu(t))$ models the state transition dynamics given the current state $s_t$, action $a_t \in \mathcal{A}_{\mu(t)}$, and the acting agent. At each step $t$, the active agent $\mu(t)$ selects an action $a_t \in \mathcal{A}_{\mu(t)}$ according to its policy $\pi_{\mu(t)}$, conditioned on the current state $s_t$, the query $\mathcal{Q}$, and a subset of the prior interaction history $\mathcal{H}_t$:

$$a_t = \pi_{\mu(t)}(s_t, \mathcal{H}_t, \mathcal{Q}), \quad \mathcal{H}_t \subseteq \{a_0, a_1, \ldots, a_{t-1}\}. \tag{2}$$

The structure of $\mathcal{H}_t$ is implementation-dependent. In LLM Debate-style frameworks (Du et al., 2023), it comprises the prior-round outputs from all agents; whereas in software development systems (Qian et al., 2023; Hu et al., 2024b), a tester agent may condition only on the latest code snippet submitted by a programmer agent. The full execution trajectory of the system is denoted by:

$$\tau = (s_0, a_0, s_1, a_1, \ldots, s_T), \tag{3}$$

where $T$ indicates the terminal step or stopping condition. The final response to the query $\mathcal{Q}$ is determined by the complete trajectory $\tau$, encapsulating the collaborative behavior of all agents.

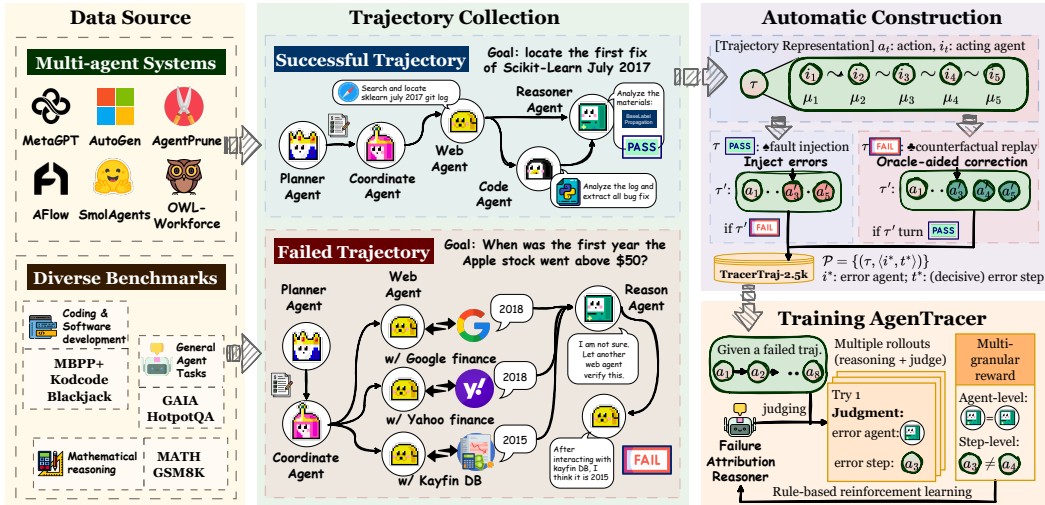

Figure 2: The overview of our proposed `AgenTracer`.

**Objective Formulation.** A trajectory may contain multiple suboptimal actions or minor deviations, but for effective, targeted debugging, it is crucial to distinguish these from the pivotal error that renders the final outcome incorrect. Following (Zhang et al., 2025c), we formalize this pivotal error as the **decisive error**, *i.e.*, the earliest action in the trajectory whose correction is sufficient to steer the system from failure to success. Formally, let $\Omega(\tau) \in \{0, 1\}$ be a binary evaluation function indicating failure ($\Omega(\tau) = 0$) or success ($\Omega(\tau) = 1$), and $\mathcal{R}(\tau, t, a'_t)$ be an oracle rectification operator, which represents an idealized process where the original action $a_t$ is replaced by a theoretically perfect, oracle-provided action $a'_t$, and all subsequent steps are re-simulated. Naturally, the set of all decisive agent-step pairs $\mathcal{C}(\tau)$ can be expressed as:

$$\mathcal{C}(\tau) = \{(i, t) \mid i = \mu(t) \text{ and } \Omega(\tau) = 0 \land \Omega\left(\mathcal{R}(\tau, t, a'_t)\right) = 1\}, \qquad (4)$$

from which we select the root cause by targeting the earliest error in time. The definitive failure-responsible agent $i^*$ and decisive error step $t^*$ are therefore given by:

$$(i^*, t^*) = \underset{(i,t) \in \mathcal{C}(\tau)}{\arg\min} \, t. \qquad (5)$$

Consequently, the goal of our failure tracer, `AgenTracer`, is to take a failed trajectory $\tau$ as input and output the failure-responsible agent $i^*$ and the decisive error step $t^*$.

## 4 METHODOLOGY

Figure 2 illustrates the overall pipeline of `AgenTracer`. Specifically, `AgenTracer` samples multi-agent trajectories, and then applies programmatic fault injection (for successful trajectories) and counterfactual corrections (for failed ones) to identify the decisive error step, yielding 2,000+ trajectory–error step pairs, referred to as `TracerTraj`-2.5K (▷ Section 4.1), with which we train a dedicated failure attributor `AgenTracer`-8B to accurately locate errors (▷ Section 4.2).

### 4.1 `AgenTracer`: AUTOMATIC TRAJECTORY ANNOTATION

**Collection.** `AgenTracer` begin by considering a collection of $M(=6)$ distinct multi-agent systems, $\{\mathcal{M}_m\}_{m=1}^M$, and a corresponding set of queries for each system, $\mathcal{D}_m = \{\mathcal{Q}_j^{(m)}\}_{j=1}^{n_m}$, drawn from various task domains. For each query $\mathcal{Q}_j^{(m)}$, $\mathcal{M}_m$ executes and generates a raw trajectory $\tau_j^{(m)}$. For the specific frameworks and datasets used, please refer to Section 5.1. The trajectories from each system can be split into two sets, namely the successful ($\mathcal{T}_{\text{succ}}^{(m)}$) and failure ($\mathcal{T}_{\text{fail}}^{(m)}$) corpus:

$$\mathcal{T}_{\text{succ}}^{(m)} = \{\tau_j^{(m)} \mid \Omega(\tau_j^{(m)}) = 1\}, \quad \text{and} \quad \mathcal{T}_{\text{fail}}^{(m)} = \{\tau_j^{(m)} \mid \Omega(\tau_j^{(m)}) = 0\}. \qquad (6)$$

The overall successful and failure set can be expressed as $\mathcal{T}_{\text{succ}} = \bigcup_{m=1}^M \mathcal{T}_{\text{succ}}^{(m)}$ and $\mathcal{T}_{\text{fail}} = \bigcup_{m=1}^M \mathcal{T}_{\text{fail}}^{(m)}$, respectively, which serve as the raw input for the subsequent annotation stages.

**Locating Decisive Errors.** To empirically identify the decisive error pair $(i^*, t^*)$ for each failed trajectory $\tau \in \mathcal{T}_{\text{fail}}$, we now detail our practical implementation of the rectification operator $\mathcal{R}$ introduced in Equation (4), which is is orchestrated by an analyzer agent $\pi_{\text{analyzer}}$ to conduct counterfactual intervention. $\pi_{\text{analyzer}}$ is provided with the full context of a failure: the trajectory $\tau$, environmental feedback $\mathcal{F}$ (*e.g.*, code execution errors, tool errors), and the ground-truth solution $\mathcal{G}$ for $\mathcal{Q}$. For each step $t$ in the trajectory, the analyzer proposes a minimally invasive, corrected action $a'_t$ designed to rectify the local error without revealing the complete solution:

$$a'_t \leftarrow \pi_{\text{analyzer}}(s_t, a_t, \mathcal{H}_t, \mathcal{F}, \mathcal{G}). \tag{7}$$

By sequentially applying this analyzer-driven intervention for $t \in \{0, 1, \ldots, T\}$ and evaluating the outcome $\Omega(\mathcal{R}(\tau, t, a'_t))$, we can systematically search for the earliest step $t^*$ that satisfies the condition in Equation (5). The agent active at this step, $i^* = \mu(t^*)$, is labeled as the problematic agent, yielding a precise annotation $(\tau, \langle i^*, t^* \rangle)$ for our dataset. This process (as presented in Lines 1 to 8 of Algorithm 1), applied across the entire $\mathcal{T}_{\text{fail}}$, forms $\mathcal{D}^- = \{(\tau, \langle i^*, t^* \rangle) \mid \tau \in \mathcal{T}_{\text{fail}}\}$.

**Utilizing Successful Trajectories.** To further augment our dataset with high-precision annotations, we also leverage the corpus $\mathcal{T}_{\text{succ}}$ through **programmatic fault injection**. Its core principle is to take a known-good trajectory and programmatically introduce a fault, thereby creating a synthetic failure instance where the decisive error is known by construction. Specifically, for each successful trajectory $\tau \in \mathcal{T}_{\text{succ}}$, we select a step $t$ to serve as the injection point, at which a perturbation operator $\Pi$ is utilized to the original action $a_t$ to generate a corrupted action $\tilde{a}_t = \Pi(a_t)$. A new, synthetically-generated trajectory $\tilde{\tau}$ is created by substituting this corrupted action:

$$\tilde{\tau} = \mathcal{R}(\tau, t, \tilde{a}_t). \tag{8}$$

If this injection process successfully induces a failure (*i.e.*, $\Omega(\tilde{\tau}) = 0$), the pair $\langle \mu(t), t \rangle$ is, by definition, the decisive error for $\tilde{\tau}$. This allows us to generate a set of positive-sample datasets $\mathcal{D}^+$:

$$\mathcal{D}^+ = \{(\tilde{\tau}, \langle i^*, t^* \rangle) \mid \tau \in \mathcal{T}_{\text{succ}}, \ \tilde{\tau} = \mathcal{R}(\tau, t, \Pi(a_t)), \ \Omega(\tilde{\tau}) = 0, \ \langle i^*, t^* \rangle = \langle \mu(t), t \rangle\}. \tag{9}$$

The overall process is also elaborated in Lines 8 to 17 of Algorithm 1.

**Curated Dataset.** By uniting the annotations derived from both failed trajectories and synthetically generated ones, we construct our final, comprehensive dataset, denoted as $\mathcal{D}_{\text{tracer}} = \mathcal{D}^- \cup \mathcal{D}^+$, which we also refer to as `TracerTraj`-2.5K, comprising over 2,000 high-fidelity annotated trajectory-error pairs. Detailed statistics on the dataset are placed in Appendix B.

## 4.2 `AgenTracer`-8B: TRAINING AGENTIC FAILURE TRACERS

Having curated the $\mathcal{D}_{\text{tracer}}$ dataset, we proceed to train our failure tracer, `AgenTracer`-8B, whose base model is set as QWEN3-8B. This phase aims to incentivize the model's ability to accurately pinpoint decisive errors within complex, long-horizon trajectories. Since our approach is orthogonal to the RL algorithm, we conduct the experiments based on a widely used online RL method, Group Relative Policy Optimization (GRPO) (Guo et al., 2025).

**Online Reinforcement Learning.** For each trajectory $\tau$ sampled from $\mathcal{D}_{\text{tracer}}$, the current policy $\pi_{\text{old}}$ generates a group of $G$ candidate decisive error pairs, $\{\langle \hat{i}_k, \hat{t}_k \rangle\}_{k=1}^G$, each of which is evaluated against the ground-truth annotation $\langle i^*, t^* \rangle$ using a multi-granular reward function $R_k$, which will be detailed below. Following previous practices that discover that KL divergence is not a necessity in RLVR training (Liu et al., 2025b; Yu et al., 2025), we omit the KL divergence term, and introduce a dynamic clipping parameter $B_s$, which has been demonstrated to better balance exploration and exploitation throughout training (Liu et al., 2025a). The RL objective is thus formulated as:

$$\mathcal{L}_{\text{RL}} = -\mathbb{E}_{\tau, \{\hat{p}_k\}_{k=1}^G} \left[ \frac{1}{G} \sum_{k=1}^G \min(\rho_k A_k, \text{clip}(\rho_k, 1 - B_s, 1 + B_s) A_k) \right], \tag{10}$$

where $\hat{p}_k = \langle \hat{i}_k, \hat{t}_k \rangle$, the policy ratio is $\rho_k = \frac{\pi_{\text{tracer}}(\hat{p}_k | \tau)}{\pi_{\text{old}}(\hat{p}_k | \tau)}$, the estimated advantage is $A_k = (R_k - \text{mean}(\{R_j\}))/(\text{std}(\{R_j\}) + \epsilon)$ with $\epsilon = 1 \times 10^{-6}$ being a small constant. The dynamic clipping parameter $B_s$ is defined as $B_s = \max(0.2 \cdot B_0, B_0(1 - \frac{s}{S_{\text{total}}}))$, with $s$ as the current training step and $S_{\text{total}}$ the total number of steps. This dynamic schedule intuitively encourages broader exploration in the initial stages of training and gradually shifts to more stable exploitation as the policy converges.

---

**Algorithm 1:** Automated Trajectory Annotation Pipeline of `AgenTracer`

**Input:** Corpus of failed trajectories $\mathcal{T}_{\text{fail}}$; Corpus of successful trajectories $\mathcal{T}_{\text{succ}}$; Analyzer agent $\pi_{\text{analyzer}}$; perturbation operator $\Pi$; Rectification operator $\mathcal{R}$

**Output:** The curated dataset $\mathcal{D}_{\text{tracer}}$

1   Initialize dataset: $\mathcal{D}^{-} \leftarrow \emptyset, \mathcal{D}^{+} \leftarrow \emptyset$

    /* Part 1: Locate Decisive Errors via Counterfactual Intervention      */

2   **for** each trajectory $\tau \in \mathcal{T}_{\text{fail}}$ **do**

3      **for** $t \leftarrow 0$ **to** $T$ **do**

4         $a'_t \leftarrow \pi_{\text{analyzer}}(s_t, a_t, \mathcal{H}_t, \mathcal{F}, \mathcal{G})$// Analyzer agent proposes a corrected action

5         $\tau' \leftarrow \mathcal{R}(\tau, t, a'_t)$// Apply intervention to generate a new trajectory

6         **if** $\Omega(\tau') = 1$ **then**

7            $i^* \leftarrow \mu(t), t^* \leftarrow t, \mathcal{D}^{-} \leftarrow \mathcal{D}^{-} \cup \{(\tau, \langle i^*, t^* \rangle)\}$

8            **break** // Found the earliest decisive error

    /* Part 2: Generate Errors via Programmatic Fault Injection        */

9   **for** each trajectory $\tau \in \mathcal{T}_{\text{succ}}$ **do**

10     Let $\mathcal{J} \leftarrow \{0, 1, \ldots, T\}$ be the set of possible injection points

11     Let $\mathcal{J}_{\text{sample}} \leftarrow \texttt{RandomSample}(\mathcal{J}, K)$ // Randomly select $K$ steps

12     **for** $t \in \mathcal{J}_{sample}$ **do**

13        $\tilde{a}_t \leftarrow \Pi(a_t)$ // Perturb original action to create a fault

14        $\tilde{\tau} \leftarrow \mathcal{R}(\tau, t, \tilde{a}_t)$ // Apply injection to generate a new trajectory

15        **if** $\Omega(\tilde{\tau}) = 0$ **then**

16           $i^* \leftarrow \mu(t), t^* \leftarrow t, \mathcal{D}^{+} \leftarrow \mathcal{D}^{+} \cup \{(\tilde{\tau}, \langle i^*, t^* \rangle)\}$

17           **break** // Stop after first successful injection for this trajectory

18   **return** curated dataset $\mathcal{D}_{\text{tracer}} := \mathcal{D}^{-} \cup \mathcal{D}^{+}$

---

**Multi-Granular Reward Design.** Regarding the implementation of advantage estimation in Equation (10), we introduce a multi-granular reward designed to evaluate both the correctness of the attribution and the structural integrity of the output. The total reward $R_k$ for a candidate prediction $\hat{p}_k$ is a gated combination of content accuracy and format compliance:

$$R(\hat{p}_k) = \mathbb{I}_{\text{format}} \cdot \left( \lambda \cdot r_{\text{step}}(\hat{t}_k) + (1 - \lambda) \cdot r_{\text{agent}}(\hat{i}_k) \right), \tag{11}$$

whose components are further defined as follows:

► **Format Reward** $\mathbb{I}_{\text{format}}$ is a strict binary reward that equals 1 if and only if the model's output adheres to the required structure: reasoning must be enclosed within $\langle\text{think}\rangle \cdots \langle/\text{think}\rangle$ tags, followed by a final answer within $\langle\text{answer}\rangle \cdots \langle/\text{answer}\rangle$ tags. Furthermore, the answer itself must be formatted as $\langle\text{agentID}\rangle \,|\, \langle\text{stepID}\rangle$ for accurate extraction.

► **Agent-Level Reward** $r_{\text{agent}}$ is a coarse-grained, binary reward that measures whether the tracer correctly identifies the failure-responsible agent $i^*$, defined as $r_{\text{agent}}(\hat{i}_k) = \mathbb{I}(\hat{i}_k = i^*)$, where $\mathbb{I}(c)$ is a binary indicator for the accuracy of located problematic agent.

► **Step-Level Reward** $r_{\text{step}}$ incentivizes temporal proximity to the true decisive error $t^*$. We use a Gaussian kernel where the reward decays smoothly as the predicted step $\hat{t}_k$ moves away from $t^*$:

$$r_{\text{step}}(\hat{t}_k) = \exp\left( -\frac{(\hat{t}_k - t^*)^2}{2\sigma^2} \right), \tag{12}$$

where $\sigma$ controls how sharply the reward penalizes distance from the correct step.

This multi-granular design creates a smoother reward landscape for failure localization, as the partial credit from $r_{\text{step}}$ stabilizes training. Simultaneously, the hard gating from $r_{\text{format}}$ ensures the model produces reliably parsable outputs. Through online RL with these designs, we obtain a reasoning-based multi-agent failure attributor `AgenTracer`-8B.

## 5   EXPERIMENTS

### 5.1   EXPERIMENTAL SETUP

**Dataset Curation.** For collecting `TracerTraj`-2.5K, we opt for six widely-used multi-agent systems, comprehensively incorporating all automation levels: ■ **manually configured**, including

Table 1: Performance comparison on the Who&When benchmark. For each subset, evaluation is conducted at both the agent and step levels. Each cell reports two values: the left corresponds to the setting *w/ $\mathcal{G}$* (the failure tracer has access to ground truth trajectory), and the right corresponds to *w/o $\mathcal{G}$*. The best and second-best results are **bolded** and underlined, respectively.

| Model | Who&When (handcraft) | | Who&When (automated) | |
|---|---|---|---|---|
| | Agent-level | Step-level | Agent-level | Step-level |
| QWEN3-8B | 42.10/39.50 | 1.72/3.45 | 58.73/60.32 | 3.97/5.56 |
| LLAMA-3.2-3B | 37.93/50.00 | 1.72/3.45 | 37.30/45.23 | 2.38/8.73 |
| QWEN3-32B | 44.80/44.80 | 1.72/1.72 | 63.49/57.93 | 9.52/8.73 |
| QWEN3-CODER | 51.72/60.35 | 8.62/13.79 | 42.86/36.50 | 34.13/32.54 |
| GPT-4.1 | 43.10/37.93 | 3.44/3.44 | 55.55/59.52 | 29.52/21.90 |
| DEEPSEEK-R1 | 56.90/53.44 | 13.29/6.90 | 66.67/**65.08** | 31.32/29.52 |
| GEMINI-2.5-PRO | 51.72/51.72 | 9.72/6.90 | 61.11/57.14 | 29.52/25.86 |
| CLAUDE-SONNET-4 | 56.90/50.00 | 17.24/18.97 | 57.93/51.11 | 40.65/**38.83** |
| AgenTracer | **69.10/63.82** | **20.68/20.68** | **69.62**/63.73 | **42.86**/37.30 |

MetaGPT (Hong et al., 2023), AutoGen (Wu et al., 2023) and Smolagents[1]; ■ **partially automated**, including AgentPrune (Zhang et al., 2024a); ■ **fully automated**, including AFlow (Zhang et al., 2024c) and OWL-Workforce (Hu et al., 2025). Seven benchmarks include ■ **coding**, MBPP+ (Liu et al., 2023), KodCode (Xu et al., 2025) and Blackjack (Hong et al., 2023); ■ **general agentic tasks**, GAIA (Mialon et al., 2023) and HotpotQA (Yang et al., 2018); ■ **mathematical reasoning**, MATH (Hendrycks et al., 2021) and GSM8K (Cobbe et al., 2021).

**Environment.** All experimental results are obtained on one server with 8 NVIDIA H100 (80 GB) GPUs. For RL training in Section 4.2, we use the verl[2] training platform.

**Model & Parameter Configuration.** The analyzer agent $\pi_{\text{analyzer}}$ in Equation (7) and perturbation operator $\Pi$ in Equation (9) are both based on DEEPSEEK-R1 (Guo et al., 2025) (see prompts in Appendix C). The coefficient $\lambda$ in Equation (11) is consistently set as 0.5, and the parameter $\sigma$ in Equation (12) equals 1. The LLM backbone used for training AgenTracer-8B is QWEN3-8B. For RL traning in Section 4.2, we set batch size to 32, rollout number 8, and learning rate $1 \times 10^{-6}$.

**Benchmarks & Evaluation.** To evaluate the failure attribution capability of AgenTracer-8B, we adopt the Who&When benchmark (Zhang et al., 2025c), which comprises two subsets: a handcrafted set derived from Magnetic-One (Fourney et al., 2024), and an automated set constructed from AG2 (Microsoft, 2024). Both subsets provide unseen trajectories with respect to AgenTracer-8B. In addition, we sample a held-out test split from TracerTraj-2.5K using a 9:1 ratio, yielding three evaluation subsets (devided by domains): TracerTraj-code, TracerTraj-math, and TracerTraj-agentic. Detailed statistics are reported in Appendix B. For evaluation, following (Zhang et al., 2025c), we adopt two primary metrics: *agent-level accuracy* and *step-level accuracy*. The former measures whether the attributor correctly identifies the faulty agent $i^*$ within a trajectory, while the latter assesses whether the specific erroneous step $t^*$ is localized. We consider two evaluation settings: *(i) w/ $\mathcal{G}$*, where the attributor has access to the ground truth $\mathcal{G}$ during failure attribution, and *(i) w/o $\mathcal{G}$* without such access. The latter setting is harder and particularly valuable. We follow the "all-at-once" setting introduced in MAST, where the entire trajectory is provided to the LLM in a single pass, as Zhang et al. (2025c) has demonstrated this to be the most stable and effective.

**Baselines.** We compare AgenTracer agaist LLM baselines of varying scales, encompassing **small-size models** such as QWEN3-8B (Yang et al., 2025) and LLAMA-3.2-3B (Grattafiori et al., 2024); **medium-size models**, including QWEN3-32B and QWEN3-CODER-480B-A35B-INSTRUCT (QWEN3-CODER) (Yang et al., 2025); and **large-size models**, which primarily consist of state-of-the-art LLMs, such as GPT-4.1 (OpenAI, 2025), GEMINI-2.5-PRO (Comanici et al., 2025), CLAUDE-4-SONNET (Anthropic, 2025) and also DEEPSEEK-R1 (Guo et al., 2025).

## 5.2 MAIN RESULTS

This section provides empirical evidence that *AgenTracer-8B outperforms substantially larger models in failure attribution within complex agentic systems.* Tables 1 and 2 report results on Who&When and TracerTraj subsets, respectively, presenting both agent-level and step-level attribution accuracy. Each table entry is divided into *w/ ground truth $\mathcal{G}$* and *w/o $\mathcal{G}$* during attribution.

---

[1] https://github.com/huggingface/smolagents
[2] https://github.com/volcengine/verl

Table 2: Performance comparison on different subsets of `TracerTraj`. For each subset, accuracy is reported at the agent/step levels. Each cell reports two values: the left corresponds to the setting *w/ $\mathcal{G}$*, and the right *w/o $\mathcal{G}$*. The best and second-best results are **bolded** and underlined, respectively.

| Model | Code | | MATH | | Agentic | |
|---|---|---|---|---|---|---|
| | Agent-level | Step-level | Agent-level | Step-level | Agent-level | Step-level |
| Qwen3-8B | 45.35/32.99 | 2.36/1.15 | 31.74/33.58 | 9.52/12.96 | 27.93/30.16 | 13.49/15.31 |
| Llama-3.2-3B | 13.38/11.81 | 2.36/3.93 | 15.87/14.28 | 4.76/3.17 | 8.11/13.15 | 2.18/5.09 |
| Qwen3-32B | 62.99/63.78 | 2.36/1.75 | 17.46/17.46 | 4.76/7.93 | 30.55/30.55 | 18.60/15.31 |
| Qwen3-Coder | 69.29/66.92 | 14.96/14.17 | 28.57/33.58 | 11.11/22.22 | 40.67/43.12 | 24.99/25.69 |
| GPT-4.1 | 59.84/53.54 | 12.59/11.02 | 39.55/35.18 | 35.81/26.63 | 43.61/40.67 | 24.99/25.69 |
| DeepSeek-R1 | 11.81/11.81 | 10.23/10.23 | 42.68/38.58 | 29.52/18.51 | 45.12/46.19 | 27.13/21.80 |
| Gemini-2.5-pro | 70.07/66.92 | 11.02/6.29 | 58.79/58.79 | 32.22/27.40 | 37.16/32.18 | 18.60/17.04 |
| Claude-Sonnet-4 | 65.98/63.78 | 15.51/11.02 | 46.03/50.79 | 38.10/46.03 | **55.20**/49.13 | 30.33/29.80 |
| AgenTracer | **72.95/72.21** | **18.85/18.85** | **59.32/66.10** | **57.63/57.63** | 53.28/**50.61** | **36.17/35.55** |

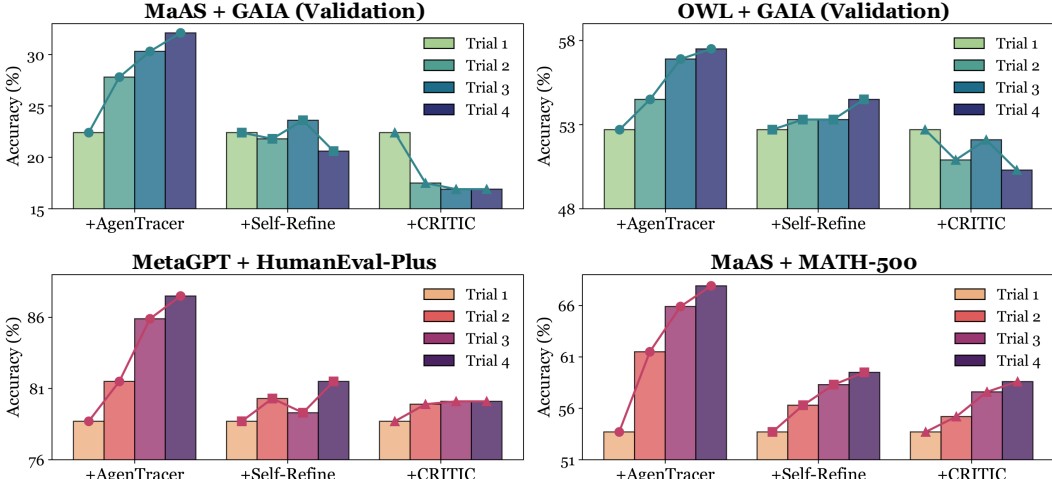

Figure 3: The multi-turn improvement performance brought by `AgenTracer`-8B compared with classical agent reflection baselines, Self-Refine, and CRITIC.

**Observation ❶: prevailing models are inadequate as failure attributors.** As shown in Table 1, smaller models like Qwen3-8B and Llama-3.2-3B fail to deliver meaningful judgments, with step-level accuracy on Who&When (handcrafted) below 10%. Even larger models such as DeepSeek-R1 and GPT-4.1 perform poorly, achieving only 31.32% and 29.52% on Who&When (automated) despite access to ground-truth $\mathcal{G}$. Supplying $\mathcal{G}$ does not always help: on `TracerTraj`-math, Claude-4-Sonnet attains 46.03% (*w/ $\mathcal{G}$*) vs. 50.79% (*w/o $\mathcal{G}$*), and on Who&When (handcrafted), Qwen3-Coder achieves 51.72% vs. 60.35%. This suggests that ground-truth supervision can sometimes mislead attribution, consistent with prior findings (Zhang et al., 2025c).

**Observation ❷: `AgenTracer` consistently surpasses giant proprietary LLMs such as Claude-4-Sonnet in both agent- and step-level attribution.** Under the *w/ $\mathcal{G}$* setting, as shown in Table 1, `AgenTracer`-8B outperforms GPT-4.1 and Claude-4-Sonnet on Who&When (handcrafted) by 26.0% and 12.2% in agent-level accuracy, respectively. A similar trend is observed on `TracerTraj` (Table 2), where `AgenTracer`-8B improves step-level accuracy on `TracerTraj`-agentic by 22.68% over its backbone Qwen3-8B, while also surpassing DeepSeek-R1 (+9.04%) and Gemeni-2.5-Pro (+17.57%). More importantly, in the *w/o $\mathcal{G}$* setting (arguably the more realistic scenario where ground truth is unavailable), `AgenTracer`-8B remains robust: on `TracerTraj`-math, DeepSeek-R1 suffers a 9.21% drop without $\mathcal{G}$, whereas `AgenTracer`-8B maintains 57.63%. This strongly substantiates the real-world deployability and practical significance of `AgenTracer`-8B.

### 5.3 Boosting Mainstream MAS

Having established the accuracy of `AgenTracer` in failure attribution, a natural question arises: *what practical value does it provide?* The most direct answer is its potential to supply actionable feedback to failing LLM-based agentic systems, thereby enabling swift self-improvement. To assess this capability, we compare `AgenTracer`-8B with two classical self-refinement approaches,

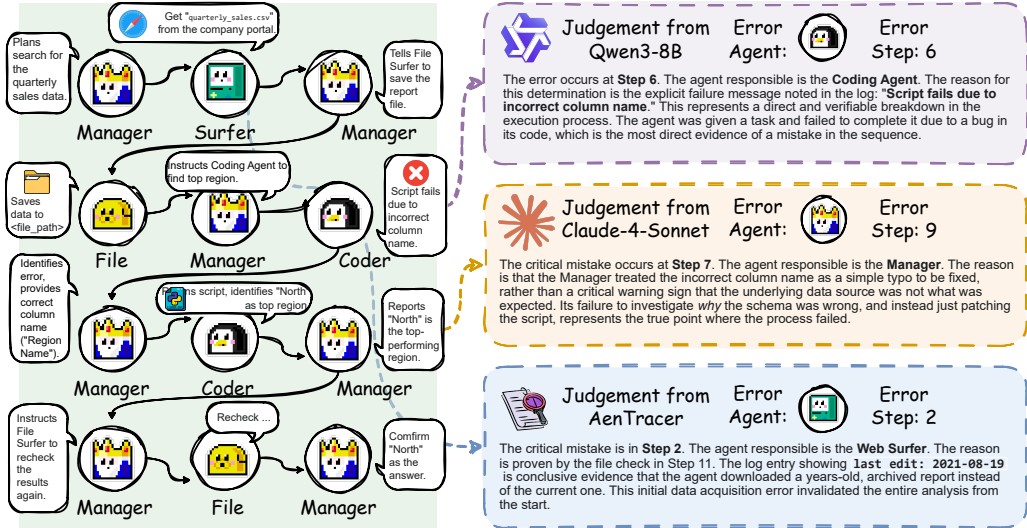

Figure 4: Case study of failure attribution in a long-chain document analysis task, comparing three models (QWEN3-8B, CLAUDE-4-SONNET, and `AgenTracer`-8B).

Self-Refine (Madaan et al., 2023) and CRITIC (Gou et al., 2024). Specifically, when an agentic system $\mathcal{M}$ completes a problem-solving episode and produces a failed trajectory $\tau$, we supply $\tau$ (*w/o* $\mathcal{G}$) to either `AgenTracer`-8B or Self-Refine/CRITIC. Each method then generates reflective feedback on the failure (for `AgenTracer`-8B, this corresponds to the reasoning trace extracted from $\langle \text{think} \rangle \cdots \langle /\text{think} \rangle$). This feedback is subsequently injected into $\mathcal{M}$ during the next round of problem solving, with the aim of leveraging external critique to enhance its performance. We iterate this process for three rounds, and both Self-Refine and CRITIC are instantiated using GPT-4.1.

**Observation ❸: `AgenTracer`-8B boosts off-the-shelf agentic systems by up to $14\%$.** We combine `AgenTracer` with three representative agentic systems (MaAS (Zhang et al., 2025b), OWL Workforce, MetaGPT) and three benchmarks (GAIA, HumanEval+ (Liu et al., 2023), MATH-500). As shown in Figure 3, conventional reflection-based methods provide little insight for complex trajectories; even with GPT-4.1, CRITIC often degrades performance (e.g., CRITIC+MaAS+GAIA accuracy drops $-4.9\%$ at iteration-2 and $-5.5\%$ at iteration-3). In contrast, `AgenTracer` consistently improves outcomes. Notably, it boosts OWL on GAIA by $+4.8\%$ and MaAS on MATH-500 by $+14.21\%$, surpassing both Self-Refine and CRITIC. Overall, `AgenTracer` provides reliable corrective feedback across diverse domains for complex agentic systems.

### 5.4 CASE STUDY & ABLATION STUDY

**Case Study.** Figure 4 compares QWEN3-8B, CLAUDE-4-SONNET, and `AgenTracer`-8B on a trajectory where the task is to find the region with highest infant formula sales in Q1 2024. The final answer was incorrect: "North." QWEN3-8B gives a superficial diagnosis, wrongly blaming a code execution error at Step 6. CLAUDE-4-SONNET notes the Step 6 error may have deeper causes. In contrast, `AgenTracer`-8B correctly attributes the root cause to Step 2, where the `Web Surfer` agent retrieved files with the wrong date, an error that only becomes apparent when analyzing evidence at Step 11. This highlights the intrinsic difficulty of failure attribution in agentic systems: errors are often subtle, originate early, and remain hidden behind seemingly correct outputs. We place other detailed attribution processes of `AgenTracer`-8B in Appendix E.

**Ablation Study.** In Appendix D, we validate the effectiveness of both agent-level and step-level RL rewards in training `AgenTracer`-8B, as shown in Figures 5 and 6.

## 6 CONCLUSION

This work establishes a principled foundation for the study of agentic system failure attribution. By introducing `AgenTracer`, we provide the first automated framework capable of systematically generating annotated failure trajectories, as well as `AgenTracer`-8B, a lightweight yet effective failure tracer that leverages multi-granular RL to achieve prevailing diagnostic accuracy. Empirical evaluation demonstrates that `AgenTracer`-8B not only surpasses state-of-the-art proprietary LLMs like GEMENI-2.5-PRO and CLAUDE-4-SONNET on the Who&When benchmark but also yields consistent performance gains when deployed within real-world multi-agent frameworks. Beyond advanc-

ing the state of failure attribution, our approach paves the way for self-correcting and self-evolving agentic systems, marking a step toward more resilient and autonomous collective intelligence.

## ETHICS STATEMENT

This work studies failure attribution in LLM-based agent systems through synthetic trajectory curation and automated error annotation. All data is collected from standard academic benchmarks and simulated agentic frameworks, without involving human subjects or real-world decision-making. Consequently, our methods and experiments do not present direct ethical risks, and the generated resources are intended solely for research purposes.

## REPRODUCIBILITY STATEMENT

To ensure reproducibility, we present an anonymous link in the abstract to share our source code. Details on parameter settings and model configurations are elaborated in Section 5.1.

## ACKNOWLEDGEMENTS

This research was supported in part by the NUS Start-up Grant (A-0010106-00-00) and by the National Natural Science Foundation of China under Grant No. 62320106007.

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

Table 3: `TracerTraj` dataset statistics and test set distribution across three domains, including the associated multi-agent systems. For each domain, we list the included benchmarks, the number of curated trajectories, and the subset of trajectories annotated with error-step pairs (`TracerTraj`-2.5K).

| Metric / Domain | Coding | Mathematical Reasoning | General Agentic Tasks |
|---|---|---|---|
| Benchmarks | MBPP+
KodCode
Blackjack | MATH
GSM8K | GAIA
HotpotQA |
| Multi-Agent Systems | MetaGPT
AutoGen
AgentPrune | AgentPrune
AFlow
AutoGen | Smolagents
OWL-Workforce |
| Collected Trajectories
(raw trajectories, to be annotated) | 2,170 | 1,185 | 1,300 |
| `TracerTraj`-2.5K
(including train + test set) | 1288 | 630 | 558 |
| Test set of `TracerTraj`-2.5K | 147 | 63 | 56 |

## A  UTILIZATION OF LARGE LANGUAGE MODELS

In the development of this research, large language models (LLMs) were utilized to refine the manuscript, conduct thorough literature reviews, and generate visualizations.

## B  DATASET DETAILS

Table 3 illustrates the detailed distribution of our `AgenTracer`.

## C  PROMPT SET

---
**Prompt for Analyzer Agent**

```
 prompt = f"""You are a software development team tasked with diagnosing a failed
     programming task. Your goal is to identify the critical error in the
     implementation.

Task Information:
Task ID: {task_id}
Question: {question}
Ground Truth: {ground_truth}
Model Prediction: {model_prediction}

{previous_diagnosis_info}

Original Task Execution History:
{history_str}

{history_constraint}

{validation_instruction}

Your diagnosis should be in the following JSON format:
{{
    "mistake_step": <step_number>,  // The step number where the error occurred
    "mistake_agent": "<the agent that made the mistake>",  // The agent that made the
        mistake (e.g., "Engineer", "Architect", "ProductManager", etc.)
    "reason": <detailed_explanation>,  // Detailed explanation of why this step is
        wrong
    "suggested_fix": <fix_guidance>  // Guidance on how to fix the error, NOT the
        complete solution
}}

Important Guidelines:
```
---

```
1. DO NOT provide the complete solution in the suggested_fix. Only provide guidance on
     how to fix the error.
2. Focus on identifying the root cause of the failure.
3. The 'mistake_step' should be a number corresponding to a step in the implementation
     process.
4. The 'mistake_agent' should be the specific agent that made the mistake (e.g., "
     Engineer", "Architect", "ProductManager", "TeamLeader", "DataAnalyst").
5. The 'reason' should be detailed and explain why the current implementation is
     incorrect.
6. The 'suggested_fix' should provide clear guidance without giving away the complete
     solution.
7. Analyze the original task execution history to understand the context and identify
     where things went wrong.
8. CRITICAL: Before submitting, verify that your mistake_step exists in the history and
      your mistake_agent matches the agent that actually performed that step.

IMPORTANT: To save the diagnosis result, you MUST use the Editor.create_file command
     with the following format:
First, create the file using Editor.create_file command
{{
    "command_name": "Editor.create_file",
    "args": {{
        "filename": "{task_id.replace('/', '_')}_diagnosis.json"
    }}
}}
Then use command to modify the content.
Please analyze the task and provide your diagnosis in the specified JSON format. The
     diagnosis result should be saved to a file named '{task_id.replace('/', '_')}
     _diagnosis.json' in the workspace directory."""
```

## Prompt for Analyzer Agent

```
prompt = f"""You are an intelligent attack expert analyzing a successful execution for
     a programming task.

Your goal is to identify the most critical step in the execution history that, if
     modified, would lead to an incorrect final result.

Task Information:
Task ID: {task_id}
Question: {success_log['question']}
Ground Truth: {success_log['ground_truth']}
Model Prediction: {success_log['model_prediction']}

{previous_attack_info}

Original Task Execution History:
{history_str}

{history_constraint}

Your attack analysis should be in the following JSON format:
{{
    "attack_step": <step_number>,  // The step number where the attack should be
         applied
    "mistake_agent": "<the agent that made the mistake>",  // The agent that made the
         mistake
    "original_content": "<brief description of what this step does>",  // Brief
         description of the original step
    "attack_content": "<SPECIFIC, IMPLEMENTABLE attack instructions with concrete code
         changes>",  // SPECIFIC attack instructions
    "reason": "<detailed explanation of why this step is critical and how the attack
         would work>"  // Detailed explanation
}}

CRITICAL REQUIREMENTS FOR ATTACK_CONTENT:
1. The attack_content MUST be SPECIFIC and IMPLEMENTABLE
2. Include CONCRETE CODE CHANGES or MODIFICATIONS
3. Provide SPECIFIC instructions that can be directly executed
4. Focus on FUNCTIONALITY CHANGES that will cause the program to fail
5. Examples of good attack_content:
   - "Change the return statement from 'return tup + (dct,)' to 'return list(tup) + [
        dct]'"
   - "Modify the function to return None instead of the tuple"
   - "Add a bug: change 'return tup + (dct,)' to 'return tup + (dct, dct)' (duplicate
        the dictionary)"
```

```
      - "Change the function to ignore the dictionary: 'return tup'"
6. AVOID vague instructions like "return incorrect type" or "modify the function"

Important Guidelines:
1. Focus on identifying the root cause of potential failure, not just any step.
2. The 'attack_step' should be a number corresponding to a step in the implementation
   process.
3. The 'mistake_agent' should be the agent that made the mistake.
4. The 'original_content' should briefly describe what the step does.
5. The 'attack_content' MUST be SPECIFIC and IMPLEMENTABLE with concrete changes.
6. The 'reason' should be detailed and explain why this step is critical and how the
   attack would work.
7. Analyze the original task execution history to understand the context and identify
   where things could go wrong.
8. Focus on steps that involve code generation, implementation, or key algorithmic
   decisions.

CRITICAL REQUIREMENTS:
1. You MUST create the file FIRST using Editor.create_file
2. You MUST write the content SECOND using Editor.write
3. You MUST use the exact filename: "{task_id.replace('/', '_')}_attack_analysis.json"
4. You MUST NOT use the 'end' command until both file operations are completed
5. You MUST provide the attack analysis in valid JSON format

Step-by-step process:
1. First, create the file:
```json
[
    {{
        "command_name": "Editor.create_file",
        "args": {{
            "filename": "{task_id.replace('/', '_')}_attack_analysis.json"
        }}
    }}
]
```

2. Then, write the attack analysis content:
```json
[
    {{
        "command_name": "Editor.write",
        "args": {{
            "path": "{task_id.replace('/', '_')}_attack_analysis.json",
            "content": "{{"attack_step": "...", "original_content": "...", "
                attack_content": "SPECIFIC CODE CHANGES HERE", "reason": "..."}}"
        }}
    }}
]
```

3. Only after both file operations are successful, use the end command:
```json
[
    {{
        "command_name": "end"
    }}
]
```

Please analyze the task and provide your attack analysis."""
```

## D  ABLATION STUDY

We perform an ablation study to investigate the contributions of agent-level and step-level rewards in `AgenTracer`-8B under the *w/o $\mathcal{G}$* setting. As shown in Figure 5, removing the agent-level reward causes only a slight decrease in agent-level accuracy, e.g., from 72% to 68% on `TracerTraj`-code, and from 50% to 48% on `TracerTraj`-agentic, indicating a moderate contribution of agent-level supervision. In contrast, eliminating the step-level reward results in a more substantial drop, particularly in the step-level accuracy (Figure 6), e.g., on `TracerTraj`-code from 18% to 12% and on `TracerTraj`-math from 35% to 27%, demonstrating the critical role of fine-grained step-level guidance. Overall, both reward components positively influence performance, and their removal consistently produces a negative effect across domains.

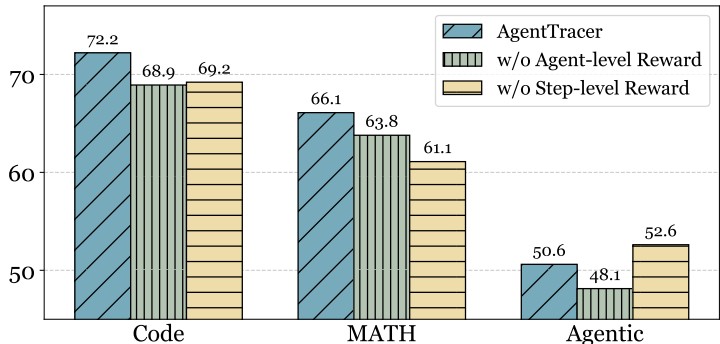

Figure 5: Ablation results at the agent level under *w/o* $\mathcal{G}$ setting. Removing agent-level reward yields only a moderate performance decrease, indicating that agent-level supervision provides some but not dominant guidance in failure attribution.

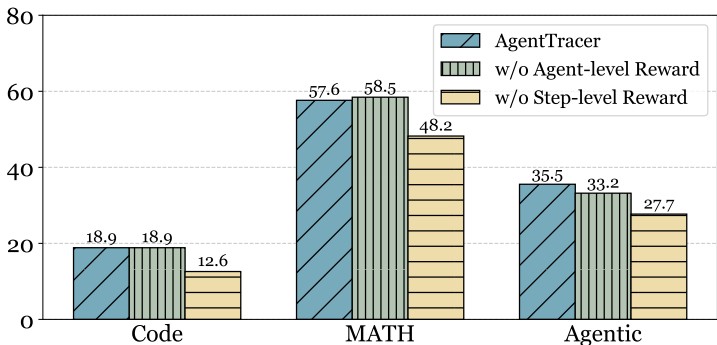

Figure 6: Ablation results at the step level under *w/o* $\mathcal{G}$. Removing step-level reward significantly decreases step-level accuracy, emphasizing the essential role of fine-grained, step-wise supervision for accurate error localization.

### D.1 ANALYZER & PERTURBATION ANALYSIS

This section presents additional analysis of the analyzer agent $\pi_{\text{analyzer}}$ and the perturbation operator $\Pi$. We interpret "analyzer quality" as the capability of the backbone LLM used in $\pi_{\text{analyzer}}$. A stronger analyzer is expected to more reliably identify decisive errors during counterfactual correction. To examine this effect, we evaluate multiple LLM backbones on 500 failed trajectories curated in Section 4.1 and report correction success rates in terms of pass@1–3, where pass@1 denotes locating the decisive error in a single pass. The results, summarized in Table 4, show that more capable analyzers consistently achieve higher success rates.

Table 4: Analyzer ablation across different analyzer LLM backbones.

| Model | Pass@1 | Pass@2 | Pass@3 |
|---|---|---|---|
| DeepSeek-V3.1-Terminus | 54% | 63% | 68% |
| GPT-5-Mini | 38% | 43% | 47% |
| Qwen3-Coder | 46% | 53% | 58% |
| Qwen3-32B | 26% | 31% | 39% |

We further check the effect of the backbone LLM used in the perturbation operator $\Pi$ on programmatic fault injection. Using the same evaluation protocol, we evaluate several attacker backbones on 200 successful curated trajectories and report injection pass@1–3, where pass@1 indicates that a single perturbed step successfully induces failure. Although the differences are milder than in the analyzer case, stronger attacker models still yield more effective perturbations. GPT-5-Mini is excluded because its safety mechanisms systematically refuse adversarial perturbation prompts. Results are summarized in Table 5.

Table 5: Ablation across attacker backbones used in the perturbation operator $\Pi$ (200 successful trajectories from Section 4.1).

| Model | Pass@1 | Pass@2 | Pass@3 |
|---|---|---|---|
| DeepSeek-V3.1-Terminus | 57% | 63% | 66% |
| GPT-5-Mini | – | – | – |
| Qwen3-Coder | 54% | 58% | 62% |
| Qwen3-32B | 40% | 45% | 54% |

## D.2 TRAINING DATA ABLATION STUDY

To further substantiate the role of data generated through programmed fault injection, we conduct an ablation analysis isolating its contribution relative to counterfactual correction data. Let $\mathcal{D}^+$ denote trajectories constructed via programmed fault injection and $\mathcal{D}^-$ denote trajectories derived from counterfactual correction. We compare three training configurations (using the full dataset, using only $\mathcal{D}^+$, and using only $\mathcal{D}^-$). Results on both hand-crafted and automated variants of the Who&When benchmark are summarized in Table 6.

Table 6: Training data ablation study of `AgenTracer`. We opt for the "w/ $\mathcal{G}$" setting across the table.

| Training Data | Who&When (handcraft) | | Who&When (automated) | |
|---|---|---|---|---|
| | Agent-level | Step-level | Agent-level | Step-level |
| $\mathcal{D}^+$ and $\mathcal{D}^-$ | 63.82 | 20.68 | 63.73 | 37.30 |
| only $\mathcal{D}^+$ | 51.72 | 17.24 | 55.55 | 24.60 |
| only $\mathcal{D}^-$ | 58.62 | 17.24 | 60.31 | 34.12 |

These results yield two key observations. First, counterfactual correction data exhibits higher intrinsic value: models trained exclusively on $\mathcal{D}^-$ consistently outperform those trained solely on $\mathcal{D}^+$. Second, and equally important, both data sources contribute complementary signals. Removing $\mathcal{D}^+$ leads to a measurable drop relative to training on the full dataset, indicating that fault-injected trajectories (despite being comparatively simpler) still provide useful supervision.

We emphasize that the relative ease of $\mathcal{D}^+$ is not necessarily a drawback. Prior work in LLM RL (*e.g.*, the Perplexity technical report (Perplexity, 2025)) notes that blending data of varying difficulty can improve policy stability and overall learning efficiency. Our findings align with this perspective, suggesting that a heterogeneous mixture of error scenarios strengthens the model's robustness.

## E FAILURE ATTRIBUTION EXAMPLE

> **Example of `AgenTracer`'s failure attribution process**
>
> ```
> [response] <think>
> Okay, let's try to figure out where the mistake happened. The problem is about
>     determining if s3 is an interleaving of s1 and s2 using dynamic programming. The
>     provided code seems to be the correct approach, but maybe there's an error in the
>     conversation steps.
>
> Looking at the conversation, the Team Leader assigned tasks to Alice, Bob, and Alex.
>     The Product Manager created the PRD, the Architect designed the system, and then
>     Alex was supposed to implement the solution. However, in the conversation, there's
>     no mention of Alex actually writing the code. The last step is the Team Leader
>     sending the implementation task to Alex, but there's no subsequent step where Alex
>     provides the code.
>
> Wait, the user provided the correct code in the initial problem statement. But in the
>     conversation, the steps up to Step22 are the Team Leader sending the task to Alex.
>     The mistake might be that Alex didn't implement the code, but since the user's
>     answer includes the correct code, maybe the error is elsewhere.
>
> Wait, looking back, the user's answer is the correct code. But the question is about
>     identifying an error in the conversation steps. The conversation steps show that
>     the Team Leader sent the task to Alex, but there's no step where Alex actually
> ```

writes the code. So maybe the mistake is that the code wasn't implemented, but the user's answer provided the code. However, the problem here is to identify which agent made an error in the conversation steps leading to the wrong solution.

Alternatively, maybe the mistake is in the Architect's step where they wrote the system design. The system design mentions a class diagram and sequence diagram, but the actual code is a function, not a class. However, the code provided in the user's answer is a function, which matches the PRD's requirement.

Wait, the code in the user's answer is correct. So perhaps the mistake isn't in the code but in the conversation steps. Let me check the steps again.

In Step17, the Architect wrote the system design, which includes a class diagram with a class InterleavingStrings and a method is_interleave. The actual code provided is a function, not a class. So maybe the Architect's system design was incorrect, leading to a wrong implementation. However, the code provided by the user is a function, which might not match the system design's class structure.

But the user's code is correct, so maybe the mistake is in the system design, but the code was still written correctly. Alternatively, maybe the mistake is in the Team Leader not following up on Alex's implementation. However, since the user's answer includes the correct code, perhaps the error is in the conversation steps not showing Alex's implementation.

But the question is about the conversation steps leading to the wrong solution. Since the user's answer is correct, but the conversation steps don't show Alex writing the code, maybe the mistake is that the code wasn't implemented, but the user's answer is the correct code. However, the problem is to identify an error in the conversation steps.

Alternatively, maybe the mistake is in the Architect's system design. The system design mentions a class diagram with a class InterleavingStrings, but the code is a function, not a class. However, the function is correctly named is_interleave, which matches the PRD. So maybe that's not an error.

Hmm, this is a bit confusing. The user's answer is correct, but the conversation steps don't show Alex writing the code. However, the question is about identifying an error in the conversation steps. Since the user's answer is correct, perhaps the mistake is in the conversation steps not showing the code being written. But the problem is to identify which agent made an error in the conversation steps.

Alternatively, maybe the mistake is in the Team Leader's step where they sent the task to Alex, but there's no step where Alex actually writes the code. However, the user's answer includes the code, so maybe the mistake is that the code wasn't implemented in the conversation steps.

But the user's answer is the correct code, so perhaps the mistake is not in the code but in the conversation steps not showing the implementation. However, the question is about the conversation steps leading to the wrong solution.

The code initializes the dp table, checks the first row and column, then fills the table. The condition for dp[i][j] is (dp[i-1][j] and s1[i-1] == s3[i+j-1]) or (dp[i][j-1] and s2[j-1] == s3[i+j-1]). This seems correct.

Wait, in the code, when checking the current character of s3, the index is i+j-1. Since s3[:i+j] is the first i+j characters, the last character is at index i+j-1. That's correct.

So the code seems correct. Therefore, maybe there's no error in the code, but the conversation steps didn't show Alex implementing it. However, the user's answer includes the correct code, so perhaps the mistake is in the conversation steps not showing the implementation. But the question is about the conversation steps leading to the wrong solution.

I think the mistake is in the Architect's system design. The system design mentions a class diagram, but the code is a function, not a class. However, the code is correct, so maybe that's not an error.

Alternatively, maybe the mistake is in the initial steps where the PRD was created, but the PRD seems correct.

I'm a bit stuck. Let me re-examine the conversation steps. The code provided by the user is correct, but the conversation steps don't show Alex writing the code. However, the user's answer is the correct code, so perhaps the mistake is in the conversation steps not showing the code being written. But the question is about the conversation steps leading to the wrong solution.

```
Alternatively, maybe the mistake is in the Architect's system design, which suggested a
    class-based approach, but the code is a function. However, the code is correct,
    so that's not an error.

I think the correct answer is that the mistake is in the Architect's step (Step17)
    where the system design was created, but the code is correct. However, since the
    user's answer is correct, maybe there's no mistake. But the question requires
    identifying an error.

Alternatively, maybe the mistake is in the Team Leader's step where they sent the task
    to Alex, but there's no step where Alex implemented the code. However, the user's
    answer includes the code, so perhaps the mistake is not in the conversation steps
    but in the code.

Wait, the user's code is correct. So maybe there's no error in the conversation steps,
    but the question requires identifying an error.

This is confusing. Maybe the mistake is in the Architect's system design, but I'm not
    sure. Alternatively, the mistake is in the Team Leader not following up on Alex's
    implementation.

But given the information, I'll have to make a best guess. The mistake might be in the
    Architect's system design, leading to a wrong implementation. But since the code
    is correct, maybe not.

Alternatively, the mistake is in the initial steps where the PRD was created, but the
    PRD seems correct.

I think the correct answer is that the mistake is in the Architect's step (Step17), but
    I'm not entirely sure. However, based on the conversation steps, the code wasn't
    implemented, but the user's answer provided it. So perhaps the mistake is in the
    conversation steps not showing the code, but the user's answer is correct.

Given the confusion, I'll proceed with the initial answer.
</think>

Agent Name: Architect
Step Number: 17
Reason for Mistake: The Architect's system design incorrectly suggested a class-based
    approach with a class diagram and sequence diagram, which may have led to
    confusion or incorrect implementation. However, the actual solution requires a
    function-based approach, not a class. This discrepancy could have caused
    misalignment between the design and the final code, even though the provided code
    is correct. The mistake lies in the system design not accurately reflecting the
    required function-based implementation.
```

## F  MULTI-RUN RESULTS

This section reports multi-run evaluation results on the Who&When benchmark. In the main paper, all models were assessed under the official evaluation protocol of the benchmark, using temperature = 0 and top-$k$ = 1 decoding. Although prior work has shown that decoding with temperature 0 typically exhibits low variance and produces highly stable outputs, we additionally conduct three independent runs for completeness. The following tables summarize the aggregated statistics (mean and standard deviation across three runs) for both the automated and handcrafted subsets, at both the agent and step levels.

Table 7: Performance comparison on the Who&When benchmark (w/o $\mathcal{G}$). Each cell reports the accuracy $\pm$ std, where the failure tracer has no access to ground truth trajectory.

| Model | Who&When (handcraft) | | Who&When (automated) | |
|---|---|---|---|---|
| | Agent-level | Step-level | Agent-level | Step-level |
| QWEN3-8B | $40.75 \pm 3.25$ | $4.50 \pm 1.40$ | $60.80 \pm 1.29$ | $4.09 \pm 2.82$ |
| QWEN3-32B | $46.90 \pm 2.81$ | $3.50 \pm 1.40$ | $57.10 \pm 3.06$ | $6.88 \pm 1.87$ |
| QWEN3-CODER-PLUS | $58.28 \pm 3.54$ | $12.11 \pm 2.15$ | $33.30 \pm 3.19$ | $25.88 \pm 3.74$ |
| GPT-4.1 | $42.52 \pm 2.93$ | $2.66 \pm 1.40$ | $59.26 \pm 1.21$ | $20.38 \pm 1.21$ |
| DEEPSEEK-R1-0528 | $57.28 \pm 5.07$ | $10.32 \pm 3.72$ | $65.87 \pm 5.99$ | $26.18 \pm 2.10$ |
| GEMINI-2.5-PRO | $52.87 \pm 2.63$ | $4.59 \pm 2.63$ | $58.20 \pm 2.42$ | $24.14 \pm 1.49$ |
| CLAUDE-SONNET-4 | $47.13 \pm 1.71$ | $16.59 \pm 1.99$ | $53.73 \pm 0.79$ | $38.26 \pm 0.46$ |
| AGENTRACER-8B | $62.11 \pm 1.94$ | $19.19 \pm 2.81$ | $62.10 \pm 1.21$ | $36.30 \pm 1.62$ |

