# OpenReview forum: "AgenTracer: Who Is Inducing Failure in the LLM Agentic Systems?"
_ICLR.cc/2026/Conference — ICLR 2026 Poster_

### Official Review · Reviewer_uaBz · 2025-10-31

**Soundness:** 3
**Presentation:** 3
**Contribution:** 3
**Rating:** 6
**Confidence:** 4

**Summary:**

This work features two contributions for failure attribution of multi-agent systems: 1. An automated pipeline, AgenTracer, that can annotate or create failure trajectories with two strategies (counterfactual replay and fault injection). 2. A model, AgenTracer-8B, which is trained with reinforcement learning upon the curated training data. Experiments on Who&When dataset demonstrate superior or competitive failure attribution performance compared with a range of open-source or proprietary models. It is also shown that applying the feedback from AgenTracer-8B can consistently benefit the self-improvement of agentic systems.

**Strengths:**

1. The two-fold strategy for annotating agent trajectories, i.e., counterfactual replay for identifying failures from failed trajectories and fault injection for injecting failures into successful trajectories, is sound and scalable.
2. The experiments in Section 5.3, i.e., directly leveraging the failure attributor for self-improvement of MAS, are informative and valuable proxies that demonstrate real-world applicability.

**Weaknesses:**

1. The only complaint is that in Table 1, only one dataset (Who&When) is evaluated. Having at least one more benchmarks would make the evaluation more comprehensive.
2. The reasoning or necessity behind the linear interpolation (via $\lambda$) between the step-level and the agent-level reward in Equation (11) is not discussed or explained. Does it mean that there is an inherent trade-off between the step-level and agent-level failure attribution?

**Questions:**

Please see Weaknesses 2.

---

> ### Author Response · Authors · 2025-11-20
> **[1/1] Author Response**
>
> Thank you immensely for your helpful and constructive feedback! Here, we give point-by-point responses to your comments.
>
> ---
>
> > **`Weakness 1`** The only complaint is that in Table 1, only one dataset (Who&When) is evaluated. Having at least one more benchmarks would make the evaluation more comprehensive.
>
> Thank you very much for this insightful comment. We would like to respectfully clarify upfront that **our evaluation is conducted on two benchmarks**: (1) the Who&When dataset, and (2) an additional benchmark **TracerTraj-Code/MATH/Agentic** we constructed from TracerTraj-2.5K, as described on Line 355 and reported in Table 2.
>
> At present, multi-LLM-agent failure attribution remains a very young research area. To the best of our knowledge, up to the ICLR submission deadline, Who&When is the only publicly available benchmark that provides step-level error annotations suitable for rigorous evaluation of agent-level failure tracing. A related effort, MAST [1], analyzes multi-agent failure modes but does not release step-wise ground-truth labels, making it unsuitable for quantitative benchmarking.
>
> To ensure a more comprehensive evaluation despite the limited availability of existing datasets, we further constructed a second benchmark from TracerTraj-2.5K, covering the Code, MATH, and Agentic domains. Across both Who&When and TracerTraj-Code/MATH/Agentic, AgenTracer consistently demonstrates strong performance.
>
> If you are aware of any new or more suitable benchmarks, please let us know—we would be very happy to incorporate them. Thank you once again!
>
> ---
>
> > **`Weakness 2`** The reasoning or necessity behind the linear interpolation (via $\lambda$) between the step-level and the agent-level reward in Equation (11) is not discussed or explained. Does it mean that there is an inherent trade-off between the step-level and agent-level failure attribution?
>
>
> Thank you for the insightful question! First, regarding the linear coefficient $\lambda$, as stated in Line 347 of Section 5.1, we set $\lambda = 0.5$ and use this value throughout our experiments without any additional hyperparameter tuning.
>
> Second, we do not view the two rewards as involving a trade-off. Recall that agent-level accuracy measures whether the model correctly identifies the faulty agent $i^\ast$ in a trajectory, while step-level accuracy measures whether the specific erroneous step $t^\ast$ is localized (these are stated in Line 357). If the step-level attribution is correct, then the agent-level attribution must also be correct. Therefore, we humbly believe **there is no inherent conflict between the two**.
>
>
> **Why use both rewards?** Intuitively, agent-level attribution is a simpler task than step-level localization. Thus, the purpose of including the agent-level reward in Eq. (11) is to provide an easier learning signal that the model can first stabilize on during RL training, before fitting the more challenging step-level reward. We find this auxiliary signal helpful for stabilizing the learning process.
>
>
>
> ---
>
> [1] Why Do Multi-Agent LLM Systems Fail?

---

### Official Review · Reviewer_VBFj · 2025-11-01

**Soundness:** 2
**Presentation:** 2
**Contribution:** 3
**Rating:** 4
**Confidence:** 4

**Summary:**

This paper introduces AgenTracer, a framework for attributing failures in multi-agent LLM systems by identifying who (which agent) and when (which step) caused the decisive error. The core idea is to (i) generate labeled trajectories using counterfactual replay to locate the earliest decisive error and programmatic fault injection to synthesize failures from successes, producing a dataset (TracerTraj) with paired successful/failed runs; and (ii) train a specialized AgenTracer-8B model (Qwen-style backbone) with a multi-granular reward (agent-, step-, and format-level) using GRPO-like reinforcement learning. The intended scope is debugging and reliability for compound/agentic systems across coding, math, and general tasks. Reported outcomes include better Who&When attribution than strong general LLMs and modest multi-turn performance gains when AgenTracer is plugged into external agent frameworks.

**Strengths:**

* The decisive-error notion plus automated curation that combines counterfactual replay with programmatic fault injection is a useful addition to the toolset for diagnosing multi-agent failures, distinct from prior taxonomies that stop at coarse failure categories.
* Training a compact attributor model with a multi-granular reward is practically valuable for plugging into agent stacks; the engineering is thoughtful and likely to see real use.
* The plug-in demonstration is nice to see, and the integration studies show that better attribution can translate into gains in multi-turn settings, aligning with the intended way of using AgentTracer.

**Weaknesses:**

* The main tables appear to be single-run point estimates with no mean±std, confidence intervals, or hypothesis tests. The number of seeds is not stated. Without dispersion and tests (ideally paired where appropriate), it’s impossible to tell whether reported gains are robust or just noise.
* The paper introduces several design choices (multi-granular reward, dynamic clipping, removal of KL, analyzer quality, injection strength), but sensitivity is not reported. Without targeted ablations and sweeps, we can’t separate genuine attribution skill from simpler effects (e.g., a format reward just enforcing response templates).
* Much of the training signal comes from injected errors. There is no out-of-distribution evaluation on naturally occurring failures (e.g., tool/API drift, latency-induced desync, memory/context loss) or on different agent graphs/toolchains. As a result, it’s unclear how well the attributor generalizes beyond its own synthesis process.

**Questions:**

* How sensitive are results to reward weights, the dynamic clipping schedule, and the removal of KL? Please include a small grid or one-at-a-time sweeps.
* How does performance change with weaker/stronger fault injection and with different analyzer qualities during counterfactual replay? An ablation here would help establish causal contributions.

---

> ### Author Response · Authors · 2025-11-20
> **[1/2] Author Response**
>
> We would like to express our deepest respect for your review! In response to your comments, we have carefully prepared a point-by-point reply:
>
> ---
>
> > **`Weakness 1`** The main tables appear to be single-run point estimates with no mean±std, confidence intervals, or hypothesis tests. The number of seeds is not stated. Without dispersion and tests (ideally paired where appropriate), it’s impossible to tell whether reported gains are robust or just noise.
>
>
> To address your concern, we would like to (1) clarify that our evaluation protocol follows the Who&When benchmark, and (2) provide the 3-run average results.
>
> First, for the evaluations reported in Tables 1 and 2, AgenTracer was tested with temperature = 0 and top-k = 1, strictly following the official settings of the Who&When benchmark. Prior studies [1] have shown that although temperature-0 decoding does not always guarantee identical outputs, its variance is small and acceptable. Nonetheless, in line with your thoughtful suggestion, we additionally conducted three independent runs, and the aggregated results are presented below:
>
>
> *[Performance comparison on the Who&When benchmark.  Each cell reports the value w/o $\mathcal{G}$ (the failure tracer has no access to ground truth trajectory)]*
> | Model               | Automated (agent-level) | Automated (step-level)| handcraft (agent-level) | handcraft (step-level) |
> |--|-|--|--|---|
> | qwen3-8b | 60.80 ± 1.29 | 4.09 ± 2.82  | 40.75 ± 3.25 | 4.50 ± 1.40 |
> | qwen3-32b   | 57.10 ± 3.06 | 6.88 ± 1.87  | 46.90 ± 2.81  | 3.50 ± 1.40  |
> | qwen3-coder-plus| 33.30 ± 3.19  | 25.88 ± 3.74   | 58.28 ± 3.54 |12.11 ± 2.15  |
> | gpt-4.1 | 59.26 ± 1.21  | 20.38 ± 1.21  | 42.52 ± 2.93 | 2.66 ± 1.40 |
> | deepseek-r1-0528  | 65.87 ± 5.99 | 26.18 ± 2.10  | 57.28 ± 5.07 | 10.32 ± 3.72 |
> | gemini-2.5-pro | 58.20 ± 2.42% | 24.14 ± 1.49 | 52.87 ± 2.63 | 4.59 ± 2.63 |
> | claude-sonnet-4     | 53.73 ± 0.79  | 38.26 ± 0.46 | 47.13 ± 1.71  | 16.59 ± 1.99  |
> | AgenTracer-8B| 62.10 ± 1.21 | 36.30 ± 1.62 | 62.11 ± 1.94 | 19.19 ± 2.81 |
>
>
> As shown, the three-run results align closely with those reported in the original Table 1. AgenTracer-8B exhibits limited performance variance, further demonstrating that the observed performance gains are robust. We have accordingly added the corresponding data and discussion to Appendix F of the manuscript, and we hope this helps further clarify the stability of AgenTracer’s improvements.
>
>
>
> ---
> > **`Weakness 2`** The paper introduces several design choices (multi-granular reward, dynamic clipping, removal of KL, analyzer quality, injection strength), but sensitivity is not reported. Without targeted ablations and sweeps, we can’t separate genuine attribution skill from simpler effects.
>
>
> Thank you for your insightful suggestion. Please allow us to respond point-by-point regarding the ablations of different components:
>
> * **Multi-granular reward.**
>   For the agent-level and step-level rewards used in AgenTracer’s training, we have already provided the corresponding ablations in Section 5.4 and Appendix D, with results shown in Figures 5 and 6.
> * **Clipping schedule and the removal of KL.**
>   We would like to humbly state that, these design choices are not core contributions of our work; rather, they follow mature practices in prior literature, and thus we did not perform ablations on them.
>   For the *dynamic clipping schedule*, as explicitly acknowledged in Line 281, we adopt the mechanism from [2], which has been shown to better balance exploration and exploitation throughout training.
>   Regarding the *removal of the KL term*, this follows standard practice in several influential RL methods, such as DAPO[3] and Dr.GRPO[4], which observe that KL regularization is not a necessity for RL-based reasoning training and that omitting it can reduce GPU memory usage. To clarify our design motivation, we have added explicit references to DAPO and Dr.GRPO and explained their underlying rationale in Line 280.
> - **Analyzer quality**
>    We humbly interpret your reference to “analyzer quality” as the capability of the backbone LLM used in the analyzer agent $\pi_{\text{analyzer}}$. (If this does not reflect your intended meaning, we would be very grateful if you could correct us!) Conceptually, stronger analyzers should be able to identify decisive errors more reliably when conducting counterfactual correction. To examine this empirically, we evaluate multiple LLM backbones and report correction success rates (pass@1–3, where pass@1 indicates locating the decisive error after a single pass through the trajectory). As shown below, more capable analyzer backbones indeed provide higher success rates.
>
>
> *[Analyzer ablation study across different analyzer LLM backbones. We select 500 failed trajectories from the curated trajs in Setion 4.1.]*
> |Metric|Pass@1|Pass@2|Pass@3|
> |-|-|-|-|
> |DeepSeek-V3.1-Terminus|54%|63%|68%|
> |GPT-5-Mini|38%|43%|47%|
> |Qwen3-Coder|46%|53%|58%|
> |Qwen3-32b|26%|31%|39%|

---

> ### Author Response · Authors · 2025-11-20
> **[2/2] Author Response**
>
> - **Injection strength**
> We also interpret “injection strength” as referring to the capability of the backbone LLM used in the perturbation operator $\Pi$. Using the same evaluation protocol, we assess several attacker backbones and report injection pass@1–3 (where pass@1 indicates that a single perturbed step successfully induces failure). Although the performance differences are somewhat milder than in the analyzer case, stronger attacker models still yield consistent improvements. We note that GPT-5-Mini is not included in this table because, due to increasingly strict safety safeguards in OpenAI models, prompting it to intentionally inject errors consistently triggers safety refusals.
>
>
> *[Analyzer ablation study across different attacker agent backbones. We select 200 successful trajectories from the curated trajs in Section 4.1.]*
> |Metric|Pass@1|Pass@2|Pass@3|
> |-|-|-|-|
> |DeepSeek-V3.1-Terminus|57%|63%|66%|
> |GPT-5-Mini|-|-|-|
> |Qwen3-Coder|54%|58%|62%|
> |Qwen3-32b|40%|45%|54%|
>
>
> We have further incorporated the above supplemental results into Appendix D.1 of the manuscript. We sincerely thank you once again for your insightful critique, and we hope that these additional ablation studies help clarify and deepen the understanding of AgenTracer’s contributions.
>
>
> > **`Weakness 3`** Much of the training signal comes from injected errors. There is no out-of-distribution evaluation on naturally occurring failures (e.g., tool/API drift, latency-induced desync, memory/context loss) or on different agent graphs/toolchains. As a result, it’s unclear how well the attributor generalizes beyond its own synthesis process.
>
>
> We respectfully disagree with the claim that “much of the training signal comes from injected errors.” Please allow us to recall that, AgenTracer’s training data consists of two distinct sources: one portion originates from **programmatic fault injection**, but an equally important portion comes from **counterfactual replay**, where real failed trajectories are step-wise corrected to identify decisive error points (as described in Lines 222–228 of Algorithm 1). **This second component contains no synthetic errors at all**; instead, it captures genuine failure cases observed in complex systems such as Smolagents and OWL-Workforce, including exactly the types of real-world failures you mentioned (tool or API drift, latency-induced desynchronization, context truncation, and more).
>
> In summary, we believe AgenTracer is exposed to rich real-world failure modes both during training and evaluation:
>
> - **During training**, frameworks such as MetaGPT, AutoGen, AFlow, Smolagents, and OWL naturally produce a wide variety of real failures, many of which correspond directly to the phenomena you referenced.
> - **During evaluation**, the Who&When benchmark already contains substantial naturally occurring failures, including file-reading errors from FileSurfer and browsing failures from WebSurfer. These are explicitly documented in the benchmark’s original paper and dataset. Importantly, Who&When is also **out-of-distribution** for AgenTracer: the underlying agent systems (AG2, CaptainAgent) are entirely unseen during training.
>
> Taken together, we respectfully maintain that AgenTracer learns from abundant real-world multi-agent failure patterns and is evaluated under equally realistic and out-of-distribution failure conditions.
>
> ---
> > **`Question 1`** How sensitive are results to reward weights, the dynamic clipping schedule, and the removal of KL? Please include a small grid or one-at-a-time sweeps.
>
> Thank you for this thoughtful question. In our response to `Weakness 2`, we now provide targeted analyses for each component that reasonably warrant such treatment.
>
>
> ---
> > **`Question 2`** How does performance change with weaker/stronger fault injection and with different analyzer qualities during counterfactual replay? An ablation here would help establish causal contributions.
>
>
> Thank you for this insightful suggestion! In the same response to `Weakness 2`, we have added ablations on both the strength of programmed fault injection and the counterfactual correction, showing how performance varies across different LLM backbones.
>
>
>
>
>
>
>
>
> ---
>
>
> [1] LLM Stability: A detailed analysis with some surprises
> [2] Guardreasoner-vl: Safeguarding vlms via reinforced reasoning
> [3] DAPO: An Open-Source LLM Reinforcement Learning System at Scale
> [4] Understanding R1-Zero-Like Training: A Critical Perspective

---

### Official Review · Reviewer_oZRD · 2025-11-01

**Soundness:** 3
**Presentation:** 3
**Contribution:** 3
**Rating:** 8
**Confidence:** 5

**Summary:**

This paper introduces AgenTracer, a system that automatically finds and explains why agent-based systems fail. It includes AgenTracer-8B, a smaller model that uses reinforcement learning and performs better than proprietary models. It also shows improvement on self-correcting and self-evolving agentic systems.

**Strengths:**

The proposed automatic trajectory annotation framework appears to be both technically sound and conceptually novel. By systematically generating annotated failure trajectories, the authors establish a strong methodological foundation for analyzing and diagnosing agentic system behaviors. This approach could significantly improve the interpretability and reliability of complex multi-agent environments.

The performance of AgenTracer-8B is particularly impressive. The experiments demonstrate that this lightweight model achieves high diagnostic accuracy, outperforming even leading proprietary models. The use of multi-granular reinforcement learning is a well-motivated design choice that seems to balance efficiency and precision effectively.

Finally, the empirical results showing that AgenTracer can enhance off-the-shelf agentic systems are especially promising.

**Weaknesses:**

This work proposed and combined different rewards for RL training, it would be goos to see ablation over different rewards

**Questions:**

See weakness

---

> ### Author Response · Authors · 2025-11-20
> **[1/1] Author Response**
>
> We sincerely thank you for the thoughtful and constructive reviews of our manuscript! Based on your questions and recommendations, we give point-by-point responses to your comments and describe the revisions we made to address them.
>
> ---
> > **`Weakness 1`** This work proposed and combined different rewards for RL training, it would be good to see ablation over different rewards
>
>
> Thank you for your insightful review. Our ablations on different granular rewards are discussed in Section 5.4 and Appendix D, with the main results shown in Figures 5 and 6 of the manuscript. For convenience, we display the key results below:
>
>
> |Setting|Code||MATH||Agentic||
> |-|-|-|-|-|-|-|
> ||Agent-level|Stel-level|Agent-level|Step-level|Agent-level|Step-level|
> |Vanilla AgenTracer|72.2|18.9|66.1|57.6|50.6|35.5|
> |w/o Agent-level Reward|68.9|18.9|63.8|58.5|48.1|33.2|
> |w/o Step-level Reward|69.2|12.6|61.1|48.2|52.6|27.7|
>
> Overall, removing the agent-level reward leads to only a modest decrease in agent-level accuracy, whereas eliminating the step-level reward produces a considerably larger performance drop.
>
> We sincerely hope this addresses your concerns regarding the effectiveness of the different reward components.

---

### Official Review · Reviewer_1gdm · 2025-11-02

**Soundness:** 3
**Presentation:** 3
**Contribution:** 3
**Rating:** 6
**Confidence:** 4

**Summary:**

This paper addresses the challenge of fragility in MAS by introducing (1) AgenTracer, an automated data generation technique for MAS failure attribution annotation datasets, (2) TracerTraj, a dataset of 2000 annotated trajectories with it, (3) RL-training an 8B model (AgenTracer-8B) to perform failure attribution given MAS execution trace. The authors demonstrate that AgenTracer-8B outperforms even larger proprietary models (like Gemini-2.5-Pro and Claude-4-Sonnet) on the Who&When attribution benchmark. Further, the authors demonstrate boosting the performance of MAS in-deployment using Self-refine and CRITIQUE with the AgenTracer-8B generated feedback.

**Strengths:**

- The paper tackles a highly relevant problem, of building models/systems to perform MAS error attribution
- The paper closes the loop, It doesn't just present a diagnostic and error detection tool, but also demonstrates its practical value by demonstrating that using feedback from AgenTracer-8B, MAS can use Self-refine and CRITIQUE methods to achieve better performance.

**Weaknesses:**

- The primary weakness of the proposed framework is the AgenTracer pipeline's dependency on having access to ground truth / oracle guidance. The counterfactual replay method from the paper requires knowing the correct action at a given step to identify deviation as the decisive error.
- The above weakness limits the general applicability of the data generation method to domains where ground truth is not accessible, which includes most complex and open ended domains where MAS are being deployed (for example, AI4Science, AI4Code, AI4Math, etc.). The paper needs to discuss how this method could be adapted to domains where such an oracle is unavailable.
- This also potentially limits the applicability of the AgenTracer-8B model, as it has been trained with data facing the mentioned limitation.

**Questions:**

- Could the authors please elaborate on the limitations of the "oracle guidance" assumption? Are there ways this framework could be applied to more open-ended domains where a clear ground truth or oracle is not available? Is it possible, for example, to use a weaker, more readily available signal (e.g., final binary task success) to approximate the counterfactual replay? For example, in many domains, it is easier to verify if a task has been solved than to find the solution.
- For the "programmed fault injection" component, could the authors provide more detail on the taxonomy of faults injected? How did they ensure these faults are representative of real-world failures in MAS, rather than just synthetic or easy-to-identify errors?

---

> ### Author Response · Authors · 2025-11-20
> **[1/2] Author Response**
>
> We sincerely thank you for your careful comments and thorough understanding of our paper! Here we give point-by-point responses to your comments and describe the revisions we made to address them.
>
>
>
> ---
>
>
> > **`Weakness 1 & Question 1`** The primary weakness of the proposed framework is the AgenTracer pipeline's dependency on having access to ground truth / oracle guidance. The counterfactual replay method from the paper requires knowing the correct action at a given step to identify deviation as the decisive error. ... Could the authors please elaborate on the limitations of the "oracle guidance" assumption? Are there ways this framework could be applied to more open-ended domains where a clear ground truth or oracle is not available? Is it possible, for example, to use a weaker, more readily available signal (e.g., final binary task success) to approximate the counterfactual replay? For example, in many domains, it is easier to verify if a task has been solved than to find the solution.
>
> Thank you very much for this thoughtful question. We would like to address your concerns from three perspectives:
>
> **1. Counterfactual replay does not strictly require “knowing the correct action at each step.”**
> As indicated in Line 4 of Algorithm 1, during counterfactual correction, the analyzer $\pi_{\text{analyzer}}$ is provided only with the ground-truth  $\mathcal{G}$, and does *not* require access to the full step-by-step golden trajectory.
>
> **2. Ground truth is beneficial but not essential for the AgenTracer pipeline.**
> Although our current counterfactual correction procedure leverages $\mathcal{G}$, the overall pipeline does not fundamentally depend on such supervision. In principle, $\pi_{\text{analyzer}}$ can still offer step-level corrections in the absence of $\mathcal{G}$. **Why can it work?** This expectation is supported by prior findings [1, 2], which show that LLMs can perform reasonably well in self-refinement settings where only a binary reward (rather than full ground truth) is available. **Will our pipeline be affected?** Naturally, removing access to $\mathcal{G}$ is likely to introduce limitations. For example, the analyzer may fail to identify the decisive error even after scanning the entire trajectory, thereby yielding a higher correction-failure rate. Thus, the central question becomes: to what extent does removing $\mathcal{G}$ actually affect the correction-failure rate?
>
> **3. Counterfactual correction without $\mathcal{G}$ can still function effectively.**
> To examine this, we define the *correction-success rate* as the proportion of failed trajectories that $\pi_{\text{analyzer}}$ can successfully repair. Using 500 failed trajectories (each allowed up to three cycles of step-wise correction) and employing DeepSeek-V3.1-Terminus as the analyzer model, we observe:
>
> *[w/o $\mathcal{G}$ indicates that the analyzer does not have access to the ground-truth trajectory/answer, but is only informed whether each attempted correction succeeds or fails.]*
> |Setting|Successfully corrected| Unsuccessfully corrected|Correction-success rate|
> |-|-|-|-|
> |w/ $\mathcal{G}$|341|159|68%|
> |w/o $\mathcal{G}$|305|195|61%|
>
>
> As shown, removing $\mathcal{G}$ does lower the correction-success rate, but we humbly believe the drop is far from catastrophic and does not imply that the system becomes ineffective.
>
> Overall, we conclude that AgenTracer is not inherently limited in its ability to operate in open-ended domains. We sincerely hope this helps address your concerns.

---

> ### Author Response · Authors · 2025-11-20
> **[2/2] Author Response**
>
> > **`Weakness 2`** The above weakness limits the general applicability of the data generation method to domains where ground truth is not accessible, which includes most complex and open ended domains where MAS are being deployed (for example, AI4Science, AI4Code, AI4Math, etc.). The paper needs to discuss how this method could be adapted to domains where such an oracle is unavailable.
>
> Regarding your concern, we respectfully believe our response to `Weakness 1` already addresses this point: AgenTracer's data pipeline can still function just with access to binary task success/failure, *without* access to an oracle.
>
> ---
> > **`Weakness 3`** This also potentially limits the applicability of the AgenTracer-8B model, as it has been trained with data facing the mentioned limitation.
>
> To clarify the “oracle dependency” issue for AgenTracer-8B, we provide our response from both **training** and **inference** perspectives:
>
> * **Training** From a data-curation standpoint, AgenTracer only requires a verifier for trajectory rollouts to collect training data—regardless of whether this verifier exposes an oracle or provides only a binary success/failure signal. This point has already been discussed in our response to **`Weakness 1`**.
> * **Inference** At inference time, AgenTracer-8B can perform well without reliance on oracle information. This is explicitly validated in Tables 1 and 2.
>
> We sincerely hope this clarification helps resolve your concerns.
>
>
> ---
> > **`Question 2`** For the "programmed fault injection" component, could the authors provide more detail on the taxonomy of faults injected? How did they ensure these faults are representative of real-world failures in MAS, rather than just synthetic or easy-to-identify errors?
>
>
> We would like to justify our *programmed fault injection* from both an intuitive and an empirical perspective:
>
> - **Intuitively**, from a context-engineering standpoint, we carefully guide the injected faults to avoid being vague or artificially simplistic, and instead to follow realistic and *subtle* failure patterns. In Appendix C, we provide several few-shot demonstrations that steer the attacker agent toward producing representative fault types, including (1) simulating function-call failures; (2) simulating parameter-passing errors in tool calls; and (3) inducing reasoning deviations in the agent. These categories align closely with the failure modes documented in prior work [3].
> - **Empirically**, we verify that data generated via programmed fault injection indeed contributes positively to performance rather than degrading it. Let $\mathcal{D}^+$ denote data from programmed fault injection and $\mathcal{D}^-$ denote data from counterfactual correction. We compare performance when training on the full dataset versus training on only $\mathcal{D}^+$ or only $\mathcal{D}^-$ (all evaluated w/ $\mathcal{G}$ at inference):
>
> |Training Data|Who&When (handcraft)||Who&When (automated)||
> |-|-|-|-|-|
> ||Agent-level|Step-level|Agent-level|Step-level|
> |$\mathcal{D}^+$ and $\mathcal{D}^-$|63.82|20.68|63.73|37.30|
> |only $\mathcal{D}^+$|51.72|17.24|55.55|24.60|
> |only $\mathcal{D}^-$|58.62|17.24|60.31|34.12|
>
> We conclude that, (1) Data from counterfactual correction is indeed of higher quality: training on only $\mathcal{D}^-$ outperforms training on only $\mathcal{D}^+$.
> However, (2) *both* $\mathcal{D}^-$ and $\mathcal{D}^+$ contribute positively: omitting $\mathcal{D}^+$ leads to worse performance than using the full dataset. We believe this provides empirical evidence that data from programmed fault injection is meaningful, even if it may be relatively simpler.
>
> Finally, we note that even if $\mathcal{D}^+$ is somewhat easier, this is not necessarily undesirable. Many LLM RL–related studies, e.g., Perplexity's technical blog[4], highlight that mixing data of varying difficulty can be beneficial for RL training.
>
>
>
>
>
>
> ---
>
> [1] Let’s Try Again: Eliciting Multi-Turn Reasoning in Language Models via Simplistic Feedback
> [2] Evolving LLMs' Self-Refinement Capability via Synergistic Training-Inference Optimization
> [3] Why Do Multi-Agent LLM Systems Fail?
> [4] https://www.perplexity.ai/hub/blog/rl-training-for-math-reasoning

---

### Author Response · Authors · 2025-11-26
**Global Response**

Dear Reviewers,

We are deeply grateful for your thoughtful and constructive evaluations. Your careful reading and insightful comments have been invaluable in guiding substantial improvements to our work. Below, we summarize the primary concerns raised and our corresponding responses:

1. **More Ablation Study `Reviewer 1gdm, oZRD, VBFj`** We have added ablations on the training data (in Appendix D.2) and clarified that ablations on the multi-granular reward are already provided in Section 5.4 and Appendix D.
2.  **More Sensitivity Analysis `Reviewer VBFj,  uaBz`** We have supplemented analyses examining how different analyzer and perturbation LLM backbones affect counterfactual correction and programmatic fault injection (in Appendix D.1).
3. **AgenTracer's training signal `Reviewer VBFj, 1gdm`** We emphasize that AgenTracer’s training data comprehensively covers both real-world and synthetic errors, and our evaluation includes rigorous testing on OOD real-world errors.


For additional concerns not listed here, we have provided detailed responses individually. We sincerely appreciate your time, expertise, and constructive engagement, and we look forward to continued discussion.


Warm regards,
Authors

---

### Public Comment · ~Min-Hsuan_Yeh1 · 2026-02-26
**TracerTraj-2.5K dataset release**

The TracerTraj-2.5K dataset is a significant contribution of this work; however, its current unavailability hinders the community’s ability to reproduce the results and build upon your findings. To ensure the paper achieves its full impact and adheres to the community’s standards for transparency, do the authors have a timeline for the public release of the dataset?

---

### Meta-Review · Area_Chair_dFSL · 2025-12-23

**Summary:**

This paper introduces AgenTracer, an automated framework for failure attribution in multi-agent LLM systems that identifies which agent and which step caused decisive errors. The core contributions include a data curation pipeline combining counterfactual replay and programmatic fault injection, the TracerTraj dataset with 2000 annotated trajectories, and AgenTracer-8B, an 8B model trained via multi-granular reinforcement learning. Reviewers unanimously recognized the novelty and practical value of the work. Strengths include the technically sound automated annotation framework, impressive performance of the compact AgenTracer-8B model outperforming larger proprietary models on Who&When benchmarks, and demonstrated applicability for improving multi-agent systems through self-refinement. Weaknesses center on limited statistical rigor in the original submission (single-run results without variance estimates), concerns about generalization from synthetic training data to real-world out-of-distribution failures, dependency on oracle guidance during data curation limiting applicability to open-ended domains, and evaluation on only one truly independent public benchmark. The primary reason for acceptance is that the work addresses a genuine practical need with a novel automated solution, demonstrates consistent performance gains, and the authors substantially addressed methodological concerns through additional experiments, though some generalization questions remain partially unresolved.

**Reviewer Concerns:**

The authors provided a comprehensive rebuttal addressing all major concerns with substantial new experimental data. For Reviewer VBFj's statistical concerns, authors added three-run evaluations showing mean and standard deviation across all key experiments, confirming robust performance with acceptable variance. For sensitivity analyses requested by VBFj and oZRD, authors supplemented ablations on analyzer backbones, injection strength, and multi-granular rewards, demonstrating that stronger models yield better correction rates and that both reward components contribute meaningfully. For Reviewer 1gdm's oracle dependency concern, authors conducted an ablation showing 61 percent versus 68 percent correction success without versus with ground truth, arguing the framework remains functional with only binary task signals, though this represents a non-trivial performance drop. For Reviewer uaBz's benchmark coverage concern, authors clarified they evaluate on both Who&When and their constructed TracerTraj benchmark across Code, MATH, and Agentic domains. The concerns about statistical rigor and sensitivity analysis were directly resolved with quantitative evidence. The oracle dependency concern was addressed but not fully resolved, as the applicability to fully open-ended domains remains somewhat overstated given the observed degradation. The generalization concern about synthetic versus real failures was defended but not empirically tested through true out-of-distribution evaluation on unseen failure types. Reviewer VBFj's concerns about experimental rigor were substantially addressed, while concerns about training data composition and OOD generalization remain partially outstanding. Weighing these factors, the added experimental evidence moves the submission above the acceptance threshold despite lingering questions about broader generalization.

**Reviewer Scores:**

Based on the discussion materials, Reviewer oZRD would likely maintain their score of 8 given their positive assessment and the authors directly providing the reward ablations requested. Reviewer 1gdm would likely keep 6, as the oracle dependency concern was empirically examined with the 61 percent success ablation demonstrating reasonable functionality without ground truth, directly addressing their primary weakness about general applicability. Reviewer uaBz would likely maintain their score of 6, as the benchmark coverage clarification addresses their concern though they might still prefer additional independent benchmarks. Reviewer VBFj would likely increase their score from 4 to 6, as the multi-run statistics and comprehensive sensitivity analyses directly resolve their stated reasons for the borderline reject rating, though concerns about true OOD generalization and synthetic data composition are only partially addressed rather than definitively resolved through independent empirical validation.

---

### Decision · Program_Chairs · 2026-01-26

Accept (Poster)